



# Interpretation of mass spectra by a Vocus proton transfer reaction mass spectrometer (PTR-MS) at an urban site: insights from gas-chromatographic pre-separation

Ying Zhang[1], Yuwei Wang[1a], Chuang Li[1], Yueyang Li[1], Sijia Yin[1], Megan S. Claflin[2], Brian M. Lerner[2], Douglas Worsnop[2], and Lin Wang[1,3,4,5,6]

[1]Department of Environmental Science and Engineering, Jiangwan Campus, Shanghai Key Laboratory of Atmospheric Particle Pollution and Prevention (LAP[3]), Fudan University, Shanghai 200438, China
[2]Aerodyne Research, Inc., Billerica, Massachusetts 01821, United States
[3]Shanghai Institute of Pollution Control and Ecological Security, Shanghai 200092, China
[4]IRDR International Center of Excellence on Risk Interconnectivity and Governance on Weather/Climate Extremes Impact and Public Health, Fudan University, Shanghai, China
[5]National Observations and Research Station for Wetland Ecosystems of the Yangtze Estuary, Shanghai, China
[6]Collaborative Innovation Center of Climate Change, Nanjing, 210023, China
[a]now at: Department of Earth and Environmental Sciences, University of Manchester, Manchester M13 9PL, United Kingdom

*Correspondence to*: Lin Wang (lin_wang@fudan.edu.cn)

**Abstract.** Volatile organic compounds (VOCs) are important atmospheric components that contribute to air pollution, but their accurate quantification by proton transfer reaction-mass spectrometry (PTR-MS) remains challenging. In this work, we coupled a gas chromatograph (GC) prior to PTR-MS and analyzed complex ambient air in urban Shanghai to speciate the PTR signal to identify which VOC species were responsible for the generation of the ions detected by PTR. We classified 176 individual PTR signals with associated compounds resolved by the GC based on whether they could be used to quantify a VOC species without pre-separation. In this classification, category I includes 45 decent signal ions that were produced from a single VOC species, and thus can be used for reliable quantification, although some of the category I ions are not the conventionally used protonated molecular ions (MH[+]). Category II includes 39 signal ions that were produced from a group of isomers, and can be used to quantify the isomeric sum, but with an increased uncertainty if a single calibration factor for one specific isomer is used to represent all structures. Category III includes 92 signal ions that were generated from more than one non-isomeric species (e.g., through protonation, fragmentation, cluster formation) and thus merely gave an upper limit of VOC concentrations. In addition, we propose, taking aromatic compounds for instance, quantification of selected VOCs with utilization of either non-MH[+] or non-Category I ions. Our results help to achieve more comprehensive species identification and reliable VOC quantification in PTR measurements.

## 1 Introduction

Volatile organic compounds (VOCs) are ubiquitous in the atmosphere and have been extensively studied and regulated due to their negative impacts on human health (Zhou et al., 2023) and air quality (Mozaffar and Zhang, 2020). Tens of thousands of



VOCs have been observed in the atmosphere (Williams and Koppmann, 2010) as a result of the enormous variations in their primary emissions from both biogenic and anthropogenic sources, and the additional complexity acquired during their

secondary transformation (Chen et al., 2019; Li et al., 2021; Schneidemesser et al., 2010). To understand the sources, fates, and environmental and health effects of VOCs, a comprehensive identification of VOCs together with accurate quantification is essential.

Gas chromatography-mass spectrometry (GC-MS), one of the most widely used techniques for VOC measurements, separates mixed VOCs through GC and detects VOCs through various MS detectors. GC-MS enables isomer-specific measurements of

VOCs, but the chromatographic separation process, together with the potential pre-concentration step, limits the time resolution of the sample analysis and thus prevents real-time measurements of VOCs (Hamilton, 2010; Helmig, 1999; Santos and Galceran, 2002). On the other hand, proton transfer reaction-mass spectrometry (PTR-MS) is an instrument with high temporal resolution to capture the rapid variations of VOCs in a real-time manner (Badjagbo et al., 2007; Blake et al., 2009; Nozière et al., 2015; Yuan et al., 2017). This, together with other advantages of PTR-MS, such as a convenient calibration, has caused

the method to be widely adopted in recent years (Li et al., 2024c; Vettikkat et al., 2023; Wang et al., 2022; Yesildagli et al., 2023) .

PTR is considered be to a soft ionization technique. The reagent ion ($H_3O^+$) can undergo proton-transfer-reactions with VOCs that have proton affinities higher than that of $H_2O$. Ideally, the collision between the reagent ion $H_3O^+$ and an analyte molecule (M) in the ion-molecule reactor (IMR) leads to the generation of a protonated molecule $MH^+$ without fragmentation as an

assumption, so that hundreds of trace VOCs can be detected simultaneously (Hansel et al., 1995; Lindinger et al., 1998). Quantitative analysis of PTR-MS measurements of VOCs using $MH^+$ requires calibration of authentic standards, but it is impractical to calibrate all VOC species detected by PTR. For uncalibrated VOCs, their mixing ratios can be calculated theoretically (Cappellin et al., 2012) because the sensitivities of VOCs in PTR-MS measurements are considered to be proportional to their rate constants, $k_{PTR}$ (Sekimoto et al., 2017; Smith and Španěl, 2011), of the corresponding proton transfer

reactions, providing an approach to estimate the quantity of VOCs that have not been explicitly calibrated for (Sekimoto and Koss, 2021).

Inter-comparisons between PTR-MS and other measurement techniques such as GC and liquid chromatograph (LC) with mass spectrometry or flame ionization detectors have been widely performed (Anderson et al., 2019; Cui et al., 2016; Dunne et al., 2018; Gouw et al., 2003a; Gouw and Warneke, 2007). Several VOCs, for example acetaldehyde, acetic acid, and isoprene

show poor agreements (Coggon et al., 2023; Cui et al., 2016; Dunne et al., 2018; Gouw et al., 2003a, b; Gouw and Warneke, 2007; Warneke et al., 2003). This observation can be attributed to multiple reasons. In the chromatographic measurement, for example, inappropriate columns and/or temperature programming lead to an incomplete elution and underreporting, and contamination of the $Na_2SO_3$ ozone trap resulted in the production of artifact aldehydes (Gouw et al., 2003a). In the PTR measurements, for example, side ion-molecule reactions including fragmentation, dehydration, and water-clustering between

M and $H_3O^+$ lead to complex product ion distributions in addition to the protonated molecular ion $MH^+$ (Romano and Hanna, 2018). The distribution between fragmentation, dehydration and water-clustering depends on the E/N ratio (where E is electric



field and N is the concentration of neutral particles) (Link et al., 2024a) . Under the condition of a low E/N, the fragmentation and dehydration can be diminished, but undesired reactions with higher order water clusters ($H_3O^+(H_2O)_n$, n≥1) that produce $[MH^+(H_2O)_n]^+$ exists. The presence of $[MH^+(H_2O)_n]^+$ ions complicates the mass spectra interpretation; $H_3O^+(H_2O)_n$ (n≥1) are

also unfavorable because the presence of higher order water clusters in the IMR alters the kinetics of the analyte ionization occurring. For example, some VOCs have lower proton affinities than $H_3O^+(H_2O)_n$ and are not easily protonated under high $H_3O^+(H_2O)_n/H_3O^+$ conditions (Gouw and Warneke, 2007; Sekimoto and Koss, 2021), causing diminished and selective detection. Under sufficiently high E/N conditions, the formation of $[MH^+(H_2O)_n]^+$ and $H_3O^+(H_2O)_n$ (n≥1) can be inhibited, but unwanted fragmentation and dehydration processes can be enhanced (Gouw and Warneke, 2007; Sekimoto and Koss, 2021).

To optimize the PTR-MS operation to minimize these unwanted processes, moderate E/N conditions are generally chosen, but neither fragmentation, nor dehydration or water clustering can be completely avoided. In addition to water, the discharge of back-flowed nitrogen and oxygen produces reagent ions $O_2^+$ and $NO^+$ in the IMR to ionize VOC molecules via other ionization pathways (e.g., charge transfer to form $M^+$ signal ions) (Link et al., 2024a) .

Fragmentation and dehydration of $MH^+$ and generation of $M^+$ lead to interferences with lower m/z ions, and formation of

$[MH^+(H_2O)_n]^+$ cluster interferes with larger m/z ions (Leglise et al., 2019; Pagonis et al., 2019). Thus, artifacts arise when measuring ambient air with complex VOC mixtures, since many ion formulas can be produced by multiple VOCs with different molecular formula (Baasandorj et al., 2015). The lack of specificity by the PTR to solely produce protonated molecular ions ($MH^+$) of the VOC molecules (M) makes it difficult to accurately quantify VOC molecules without further analysis or employment of complementary analytical methods.

One way to study the possible interferences incurred during PTR-MS measurements is to measure standards (Ambrose et al., 2010; Aprea et al., 2007; Brown et al., 2010; Buhr et al., 2002; Leglise et al., 2019; Li et al., 2024a; Romano and Hanna, 2018; Tani et al., 2003). With the elucidation of full product ion distributions generated by an authentic VOC standard in the PTR, the user can determine whether this VOC will interfere with m/z values that are used to quantify other VOCs. For example, previous studies show that pentanal ($C_5H_{10}O$) (Li et al., 2024a) and octanal ($C_8H_{16}O$) (Buhr et al., 2002) undergo fragmentation

in the IMR to generate $C_5H_9^+$ signals that interfere with the measurement of isoprene ($C_5H_8$), and that ethyl acetate ($C_4H_4O_2$) generates $C_2H_5O_2^+$ signals that interfere with the measurement of acetic acid ($C_2H_4O_2$) (Aprea et al., 2007). Although libraries for reference are available (Pagonis et al., 2019; Yáñez-Serrano et al., 2021; Yuan et al., 2017), it is impractical to quantitatively account for all these potential interferences, given the number of VOCs that can be simultaneously ionized and detected by PTR-MS and that the interferences are dependent on the environment.

Another approach is to pre-separate VOCs via chromatographic techniques, for instance GC, prior to their ionization in the PTR reactor (Coggon et al., 2023; Gouw et al., 2003b; Gouw and Warneke, 2007; Link et al., 2024a; Warneke et al., 2003). In situ GC pre-separation properly characterizes relative contributions of different VOC species to a PTR signal of interest in an ambient measurement. A key assumption to this approach is that the species detected by PTR are not lost in the pre-concentration and separation processes of the GC, i.e., the GC chromatogram should separate and elute all species that can be

detected by PTR; and preferably these species can be identified unambiguously.



PTR-MS coupled with GC has been deployed to analyze ambient atmospheric samples from the city of Utrecht in The Netherlands (Gouw et al., 2003b), the remote Sonnblick Observatory in Austria (Gouw et al., 2003b), the city of Boulder, Colorado in the United States (Warneke et al., 2003), a forest in northern Wisconsin in the United States (Vermeuel et al., 2023), and the city of Las Vegas in the United States (Coggon et al., 2023). GC chromatograms of several key PTR signals

were investigated, showing varying extents of disturbance in different locations and seasons (Coggon et al., 2023; Gouw et al., 2003b; Warneke et al., 2003). The investigated PTR signals and their corresponding potential identities are summarized in Table 1. These studies have predominately presented measurements from relatively clean sites compared to the typical air quality in Shanghai, which will be the focus of our study. Since VOC interferences in more polluted air samples could be much more severe, there is an urgent demand to expand our knowledge on interferences to the full PTR-MS spectra in new

environments and to establish a method to derive accurate VOC concentrations from PTR-MS measurements.



**Table 1** Attribution of PTR signals to atmospheric species confirmed with the combination of GC and PTR-MS.

| m/z[a] | Signal ion | Main VOC identity | Interferences[b,c,d] | | | | |
|---|---|---|---|---|---|---|---|
| | | | Utrecht (Gouw et al., 2003b) | Sonnblick (Gouw et al., 2003b) | Boulder (Warneke et al., 2003) | Wisconsin(Vermeuel et al., 2023) | Las Vegas (Coggon et al., 2023) |
| 33 | $CH_4OH^+$ | methanol | NI | NI | NI | NR | NI |
| 42 | $CH_3CNH^+$ | acetonitrile | NI | NI | NI | NR | NR |
| 45 | $C_2H_4OH^+$ | acetaldehyde | NI | UI | UI | NR | ethanol |
| 59 | $C_3H_6OH^+$ | acetone | propanal | NR | propanal | NR | propanal |
| 63 | $C_2H_6SH^+$ | dimethyl sulfide | NR | NR | NR | NI | NR |
| 69 | $C_5H_8H^+$ | isoprene | methylbutanals, pentenols | methylbutanals, pentenols | NR | heptanal, 1-nonene, octanal, and nonanal | methylbutanals, pentanal, octanal, and nonanal. |
| 71 | $C_4H_6OH^+$ | C4 carbonyls | NR | NR | NR | NR | NI |
| 79 | $C_6H_6H^+$ | benzene | NI | ethylbenzene | NI | NR | ethylbenzene benzaldehyde |
| 93 | $C_7H_8H^+$ | toluene | NI | NI | NI | NR | ethyl-methyl-benzenes |
| 105 | $C_8H_8H^+$ | styrene | NI | NR | NI | NR | NR |
| 107 | $C_8H_{10}H^+$ $C_7H_7O^+$ | C8-aromatics benzaldehyde | NI | NI | NI | NR | NI |
| 121 | $C_9H_{12}H^+$ | C9-aromatics | NI | NI | NI | NR | NI |
| 137 | $C_{10}H_{16}H^+$ | monoterpenes | NR | NR | NR | NI | NR |

Notes:

[a] PTR-MS was in a unit mass resolution (UMR) in the measurement launched in Utrecht, Sonnblick, and Boulder, and was in
a high resolution in the measurement launched in Wisconsin and Las Vegas.

[b] NR stands for "not reported".

[c] NI stands for "no interference".

[d] UI stands for "unknown interference".

In this study, we coupled an online GC equipped with thermal desorption preconcentration and two parallel chromatographic columns to a Vocus PTR-MS, and measured ambient air with a complex VOC composition at an urban site in Shanghai.



Through application of three VOC measurement modes (1) direct PTR measurements that analyze ambient air in a real-time manner (RT-PTR), (2) PTR measurements of eluted VOCs that were sampled and separated by the GC system (GC-PTR-MS), and (3) EI (electron impact) -MS measurements of eluted VOCs that were sampled and separated by the GC system (GC-EI-
MS), we established a reference table for compound identification i.e., assigning individual PTR signals to contributing compounds. Quantitative inter-comparisons between GC-PTR-MS and RT-PTR-MS were also performed to quantify the extent of interferences. Methods for appropriate quantification and correction for selected PTR signals, taking aromatic compounds as examples, were proposed.

## 2 Materials and methods

### 2.1 Measurements site

VOC measurements were conducted from 24 January to 28 February, 2022 on the rooftop of the Environmental Science Building (31.34°N, 121.52°E) at the Jiangwan campus of Fudan university, in urban Shanghai, China (Fig. S1). The site is surrounded by residential dwelling and a few industrial enterprises, and characterized by strong anthropogenic emissions (Abudumutailifu et al., 2024; Zhang et al., 2024). Note that the instrument was under maintenance from 20:00 on 11 February
2022 to 20:00 on 16 February 2022.

### 2.2 Instrument description and data acquisition

Measurements were performed in cycles that lasted one hour with the switch between three detection modes as shown in Fig. 1: (1) RT-PTR (brown): real-time measurements of ambient air using a Vocus PTR-MS, (2) GC-PTR (green): GC combined with Vocus PTR-MS, and (3) GC-EI-MS (blue): GC combined with EI-ToF-MS.
The GC system (Aerodyne Research) is equipped with two separation channels, i.e., Ch1 and Ch2. Overall, for this study, the GC system was optimized to resolve VOC and OVOCs in the C5–C15 n-alkane volatility range. Ch1 utilizes a Rxi-624 column (30 m length × 0.25 mm inner diameter, 1.4 μm film thickness, Restek, USA) that is suitable for non- to mid-polarity VOCs including hydrocarbons, oxygenates, and nitrogen- and sulfur-containing compounds. Ch2 is equipped with an MXT-WAX column (30 m length × 0.25 mm inner diameter, 0.25 μm film thickness, Restek, USA) that is suitable for separation of
hydrocarbons with higher carbon numbers and VOCs with higher polarities. The two-channel GC has an integrated two-stage thermal desorption preconcentration system (TDPCs), similar to the systems described by Claflin et al. (2020) and Vermeuel et al. (2023). When the GC system collects a sample, the sample first passes through an oxidant trap that contains activated sodium sulfite to minimize the impact of artifact-generating oxidants, like ozone, on the preconcentration steps. After the oxidant trap, the sample is split to two separate channels for preconcentration, where only Ch1 is equipped with a water trap
to remove excess water to avoid condensation in the preconcentration steps. For both Ch1 and Ch2, the sample is initially preconcentrated onto multi-bed sample traps (Markes International, Universal 1000, C3-BAXX-5070 glass tube). Following the collection onto the sample traps, the system goes through a post-collection water purge for 2 min by forward-flowing dry



gas (ultra-high purity $N_2$) through the traps. The collected sample is then thermally desorbed from the sample traps to transfer the sample to the second stage of the preconcentration system, multi-bed focusing traps (Markes International, U-T15ATA-2S

cold trap). After this second preconcentration event, each focus trap is flash heated to transfer the sample to the head of that channel's designated column.

The temperature profiles of the sample traps and the focus traps in the GC system in one typical cycle are also shown in Fig. 1. Taking Ch1 for instance, the sample traps were flushed with a high-purity helium gas at 20 cm$^3$ min$^{-1}$ (sccm) and at the same time heated, i.e., at 570 s for EI-MS detection and at 2370 s for PTR-MS detection, respectively in the cycle, to fully

desorb the captured VOCs. The sample trap heating initially ramped from 30 ℃ to 150 ℃ at a rate of 12 ℃/sec, and then from 150 ℃ to 300 ℃ at a rate of 2.5 ℃/sec. The sample traps were then held at 300 ℃ for 60 seconds and then cooled to 30 ℃ within 300 seconds. The desorbed organic molecules were transported using the same 20 sccm helium as a carrier gas to the focus traps where they were further pre-concentrated. The focus traps were flash heated to achieve a discrete thermal desorption of captured VOCs. Taking Ch1 for example, the heating processes started at 1075 s for EI-MS detection and at 2875 s for

PTR-MS detection, respectively in the cycle. The focus traps were heated from 30 ℃ to 300 ℃ within 10 second, and then held at 300 ℃ for 30 second and then cooled to 30 ℃ to concentrate collected organics onto the head of the GC columns. At the beginning of every half hour (0–300 s and 1800–2100 s in the one-hour cycle), the focus traps underwent a second heating process as described above as a precautionary cleaning procedure to remove VOCs that might remain in the previous trapping process (e.g. low-volatility species outside of the analytical range).

The temperature profiles of the two columns are also shown in Fig. 1. The two chromatographic columns, housed in separate ovens, underwent a similar temperature program after the focus traps cooled down to 30 ℃. The temperature program consisted of four phases: initially from 35 ℃ to 100 ℃ at a rate of 39 ℃/min, then from 100 ℃ to 150 ℃ at a rate of 15 ℃/min and from 150 ℃ to 220 ℃ at a rate of 30 ℃/min, and lastly held at 220 ℃ for 60 seconds for Ch1 and for 150 seconds for Ch2, respectively. The columns were cooled down in 150 seconds and kept at 35 ℃ until the next heating process.






**Figure 1: Instrument setup and temperature profiles for VOC measurements, which were switched among RT-PTR, GC-PTR, and GC-EI-TOF modes. BG-CB-BG stands for background (2 min)-calibration (2 min)-background (4 min). TDPC stands for the thermal desorption preconcentration system.**

In each instrument cycle (Fig. 1), a 2-minute background measurement was performed for the RT-PTR mode, followed by a

2-minute calibration and then a 4-minute introduction of zero gas to remove excess calibrants in the flow path. Then, PTR-MS

measured ambient air in a real-time manner for 22 minutes (brown, RT-PTR). In GC-PTR and GC-EI-MS measurements,

~760 standard cubic centimeter ($cm^{-3}$) of ambient air was sampled for 500 seconds every half hour (dark green and dark blue),

followed by preconcentration in the TDPC (described above), separated through GC columns, and then introduced into the

PTR-MS (grey-green) and the EI-MS (grey-blue) detectors alternatively. The GC collected sample for PTR-MS detection

(dark green) starting at 1225 s for 500 s in a given cycle. PTR-MS detection for GC eluates (grey-green) started at 3025 s for



Ch1, and at 2265 s for Ch2. The chromatograms were 500 s and 600 s long for Ch1 and Ch2, respectively. For EI-MS detection, the GC collected sample (dark blue) starting at 3025 s for 500 s. EI-MS detection for GC eluates (grey-blue) started at 1225 s for Ch1, and started at 465 s for Ch2. Note that during the sampling for the GC-PTR mode, EI-MS was detecting GC eluates
from Ch1, while the RT-PTR detection was running simultaneously.

A total of 1170 ambient air samples for each of Ch1 and Ch2 were collected, pre-concentrated and separated by GC, and then transferred alternately to PTR-MS and EI-MS for detection. One background check, one VOC calibration, and one residual removal for GC-PTR and GC-EI-MS measurements are performed every 26 cycles, i.e., 22 normal cycles followed by one cycle with zero air samples for GC-PTR and GC-EI-MS, one cycle with authentic VOC standards for GC-PTR and GC-EI-
MS, and again two cycles with zero air samples for GC-PTR and GC-EI-MS to remove residual calibrants.

The $H_3O^+$ ion source for the Vocus PTR-MS (Tofwerk AG (Krechmer et al., 2018)) was supplied with a 20 sccm at standard temperate and pressure (STP) flow of water vapor. The focusing IMR was operated at 100 ℃, at 2 mbar with 585 V for the axial voltage and 450 V for the radial frequency (RF) voltage at a frequency of 1.5 MHz, giving a stable $C_{10}H_{17}^+$ signal-to-all signal ratio of 0.422 for α-pinene (see Fig. S2 for detail), suggesting a stable E/N of ~130 Td (Materić et al., 2017).

The EI-TOF-MS (Tofwerk AG) used in this study is described in detail elsewhere (Obersteiner et al., 2015). The ionizer temperature was maintained at 280℃, the ionization energy was set at 70 eV, and the filament emission current was 0.2 mA.

## 2.3 Data analysis

GC-EI-MS chromatograms were used to identify VOCs in the ambient atmosphere. The measured EI mass spectrum of a chromatographic    peak    was    compared    with    standard    EI    mass    spectra    in    the    NIST    database
(https://webbook.nist.gov/chemistry/). The identification was verified together with the comparison between the measured retention time and the estimated retention time based on the Kovat's number (Dool and Kratz, 1963) queried in the NIST library with columns that have a similar polarity.

The Vocus PTR-MS was characterized with a mass resolution (full width at half maximum) of ∼ 9000 for $C_8H_{10}H^+$ (m/z, 107.0855 Th) during the measurement, allowing assignments of an ion formula to a detected PTR mass-to-charge ratio with a
deviation less than 2 ppm. Representative high-resolution fittings at ~59 Th, ~69 Th, ~79 Th, and ~107 Th are shown in Fig. S3.

After GC-EI-MS confirmation of a species, RT-PTR and GC-PTR were used for quantitative analysis. To compare PTR signals between RT-PTR and GC-PTR, the RT-PTR signals (in counts per second (cps)) that coincided with the GC-PTR sampling (500 seconds) were averaged, whereas the signals for GC eluates detected by the PTR were integrated over the GC peak elution
time to obtain total counts, then divided by 500 seconds to obtain a signal that is comparable with the RT-PTR signal (Claflin and Brian, 2023; Link et al., 2024b). The GC-PTR signal was also normalized based on the sampling volumes of GC-PTR and RT-PTR measurements.





## 3 Results and Discussion

### 3.1 Overview of PTR mass spectra

A total of 239 high-resolution PTR signals were detected in RT-PTR measurements and assigned with ion formula. Kendrick mass defects (Hughey et al., 2001) of these 239 PTR signals are shown in Fig. 2, sized by the campaign-average values of their signals in our RT-PTR measurements. The chromatograms of these 239 signals in the GC-PTR measurement were screened in all the 1170 ambient air samples. Together with 6 reagent ions ($H_3O^+$, $H_5O_2^+$, $H_7O_3^+$, $H_9O_4^+$, $O_2^{+•}$ and $NO^+$), 57 signals were absent in GC-PTR chromatograms in both channels (shown by solid gray circles in Fig. 2a), indicating that they were detected

in RT-PTR measurements but their precursor VOCs did not elute in either of the two channels of GC system. As listed in Table S1, these PTR signal ions include reagent ions, $NO_2^+$, some $C_xH_y^+$ ions that have more than seven carbon atoms, and some $C_xH_yO_z^+$ ions that have large O or/and C number. $NO_2^+$ and $C_xH_yO_z^+$ ions with large O number could be produced, respectively, by PANs (peroxyacetyl nitrate) (Yuan et al., 2017), and multifunctional oxygenated species that are generally difficult to be analyzed by GC. $C_xH_y^+$ and $C_xH_yO_z^+$ ions with large C number are likely produced, respectively, by low volitivity unsaturated

hydrocarbons and OVOCs with long carbon chains that are generally beyond our choice of GC columns and heating programs. The remaining 176 signal ions were observed in chromatograms for at least one GC channel of the GC-PTR configuration. The relative difference between signals measured by RT-PTR and GC-PTR is defined as follows, where $[PTR]_{RT,Sig}$ is the averaged RT-PTR signal (in cps) that coincided with the GC-PTR sampling (500 seconds), and $[PTR]_{GC,Sig}$ is the processed GC-PTR signal by integrating the entire chromatogram of a given ion to obtain total counts and then dividing by 500 seconds.

Also, taking into account the sampling volumes of GC-PTR and RT-PTR modes, the $[PTR]_{GC,Sig}$ was normalized.

$$Relative\ difference\ (\%) = \frac{([PTR]_{RT,Sig} - [PTR]_{GC,Sig})}{[PTR]_{RT,Sig}} \quad (Eq.1)$$

In Fig. 2b for Ch1 and Fig. 2c for Ch2, the color donates the average relative difference between RT-PTR and GC-PTR samples throughout the campaign. Positive relative differences (red circles), i.e., larger RT-PTR signals, are believed to come from uncertainties and loss of VOCs in the GC system. The number of signals that had a relative difference between 0% and 10%

was 59 in Ch1 and 97 in Ch2 respectively, with an overlap of 37. Negative relative differences (blue circles), i.e., larger GC-PTR signals, come from instrument uncertainties and a potential slight aldehydes production from the ozone reaction in the GC system (Vermeuel et al., 2023). The number of signals that had a relative difference between -10% and 0% was 34 in Ch1 and 42 in Ch2, respectively, with an overlap of 24. There were 78 signals had a relative difference between -10% and 10% in both Ch1 and Ch2. The number of signals that have a positive relative difference larger than 10% is 83 in Ch1 and 37 in Ch2,

respectively, with an overlap of 22. These 22 PTR signals, listed in Table S2, were characterized with a relatively large uncertainty in both GC channels. Most of them are $C_xH_y^+$ ions (x>7), and $C_xH_yO_z^+$ ions that have a large O or/and C number as discussed earlier. A combination of two GC channels could provide a more complete information for such an ion, for example $C_5H_9^+$ as discussed in the following section. Thus, by excluding the 22 signals that were not well characterized by both GC channels from the 176 PTR signals with chromatographic peaks, we focused on the remaining 154 had a -10% to 10%





relative difference in at least one GC channel. Consistent with characteristics of GC column Rxi-624 in Ch1 and MXT-WAX in Ch2 and the heating programs, Ch1 showed a good consistency with RT-PTR results for low m/z PTR signals such as $C_2H_5O^+$ and $C_4H_9^+$ (normally assigned to acetaldehyde and butylenes, respectively), and Ch2 showed a better performance for high m/z PTR signals such as $C_8H_9O^+$ and $C_{10}H_{15}^+$ (normally assigned to acetophenone and $C_{10}H_{14}$ aromatics, respectively). The combination of GC Ch1 and Ch2 helps to achieve measurements of more VOC species.


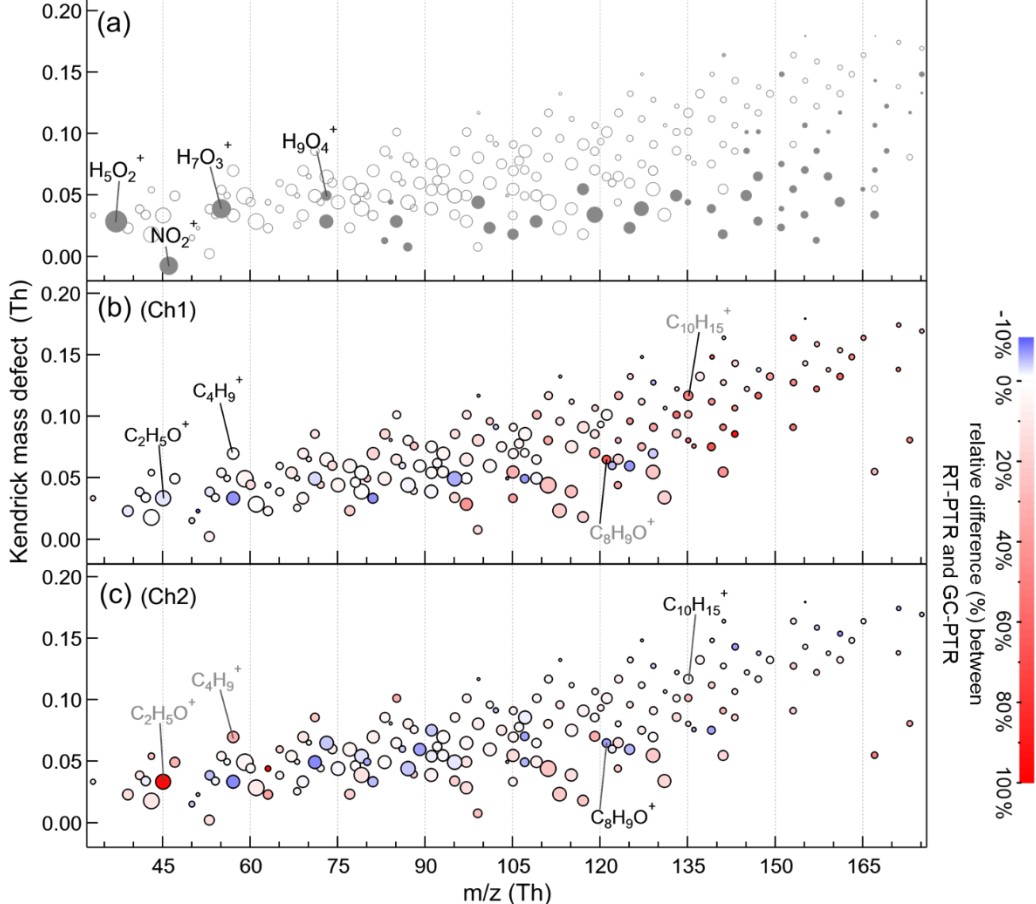

**Figure 2: Kendrick mass defects of PTR signals, sized with average values of RT-PTR signals in our measurement. 63 signals detected in RT-PTR but did not eluate in GC-PTR chromatograms in either of the two GC channels are shown in solid gray circles in (a). Also shown are 176 PTR signals that have eluted in at least one GC channel, colored with the campaign-average relative differences**
**between RT-PTR and GC-PTR throughout the measurement in (b) Ch1 and (c) Ch2, respectively.**

### 3.2 Attribution of PTR signals to VOCs

The chromatographic peaks in the GC-EI-MS that have similar retention times and peak shapes with those in the GC-PTR are located and identified. Figure 3 shows the GC-PTR chromatograms of four representative PTR signals of (A) 59.0491 Th, (B)



107.0855 Th, (C) 79.0542 Th, and (D) 69.0699 Th in both channels in the samples that were collected from 16:26:46 to

16:35:07, 19 February 2022, denoting VOCs that produce PTR signal ions $C_3H_7O^+$, $C_8H_{11}^+$, $C_6H_7^+$, and $C_5H_9^+$, respectively. High-resolution fittings and ion formula assignments are provided in Fig. S3. Identified eluates are numbered from a1 to d5 in the GC-PTR chromatogram and listed in detail in Table S3. Peaks not labeled in Fig. 3 and not listed in Table S3 are not assigned with a VOC identity.

The GC eluates that generated $C_3H_7O^+$ (a1 and a2) were identified to be acetone ($CH_3COCH_3$) in both channels. Isomers of

$C_8H_{10}$ including xylenes and ethylbenzene (b1-b7) were observed to produce the $C_8H_{11}^+$ signal as evidenced in both channels. Co-elution of m- and p-xylenes using non-polar columns (like the Rxi-624 employed here for Ch1) is a known behavior, while polar columns (like the MXT-WAX employed for Ch2) are able to separate all four of the $C_8$-aromatic isomers, as shown by the appearance of four elution peaks on Ch2 and only three elution peaks on Ch1. Authentic o/m/p-xylenes and ethylbenzene were analyzed during the GC-PTR calibration to confirm the aforementioned identification. Eluted benzene (c1 and c8),

ethylbenzene (c2 and c9), xylenes (c3, c4, c10, and c12), isopropyl-benzene (c5 and c11), n-propyl-benzene (c6 and c13) and benzaldehyde (c7 and c14) produced the $C_6H_7^+$ signal in both channels, due to fragmentation of the larger aromatic species in the IMR. $C_5H_9^+$ was produced by many identified and unidentified VOC species including isoprene (d1, Ch1), octanal (d2, Ch1 and d3, Ch2), nonanal (d4, Ch2), decanal (d5, Ch2). The $C_5H_9^+$ chromatographic peaks labeled with d-NI in Fig. 3d in Ch1 and Ch2 were identified as the same VOC species because of their identical signal values throughout the measurement

period. The specific identity was not confirmed because, as shown in Fig. S4, its co-elution with several high-abundance C5-OVOCs in both Ch1 and Ch2 during the whole campaign makes isolating its EI mass spectra and subsequent comparison with the NIST database difficult. d-NI had a PTR peak only at m/z values corresponding to $C_5H_9^+$, unlike other carbonyl compounds that would produce $MH^+$, $[M+H_2O]^+$, and $[M-H_2O]^+$ in PTR measurements (Buhr et al., 2002; Li et al., 2024a; Pagonis et al., 2019; Romano and Hanna, 2018; Warneke et al., 2003).




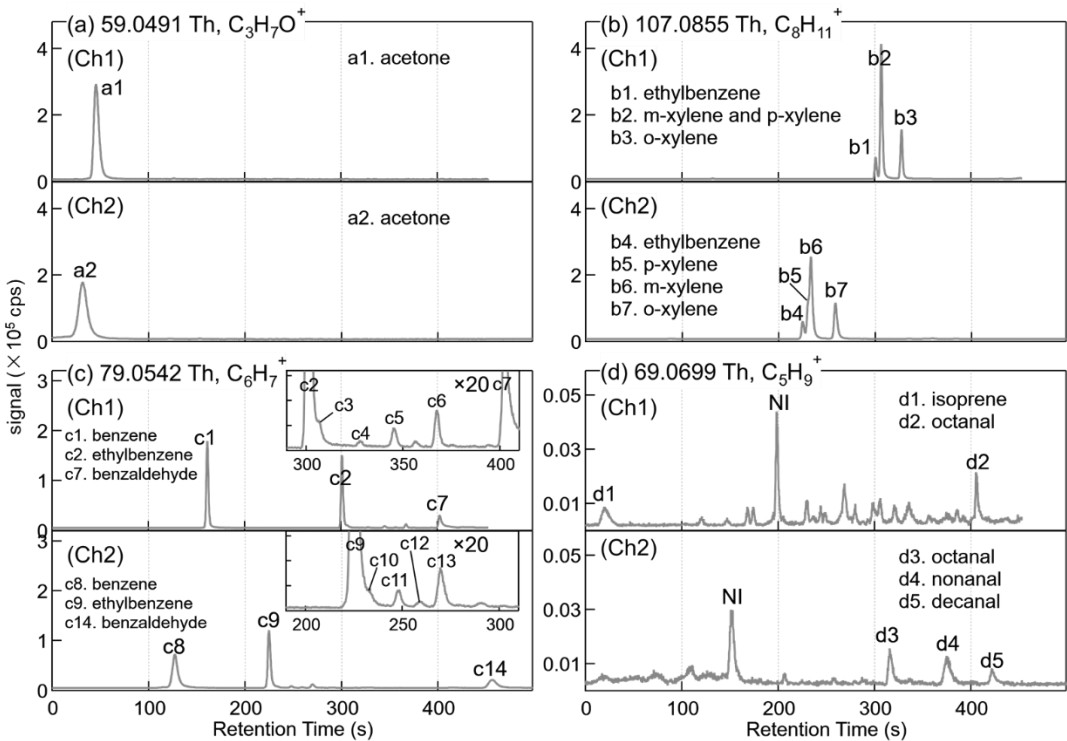

**Figure 3: GC-PTR chromatograms of PTR signals at m/z of (a) 59.0491 Th ($C_3H_7O^+$), (b) 107.0855 Th ($C_8H_{11}^+$), (c) 79.0542 Th ($C_6H_7^+$), and (d) 69.0699 Th ($C_5H_9^+$), respectively, sampled from 16:26:46 to 16:35:07 on 19 February, 2022. NI stands for "not identified".**


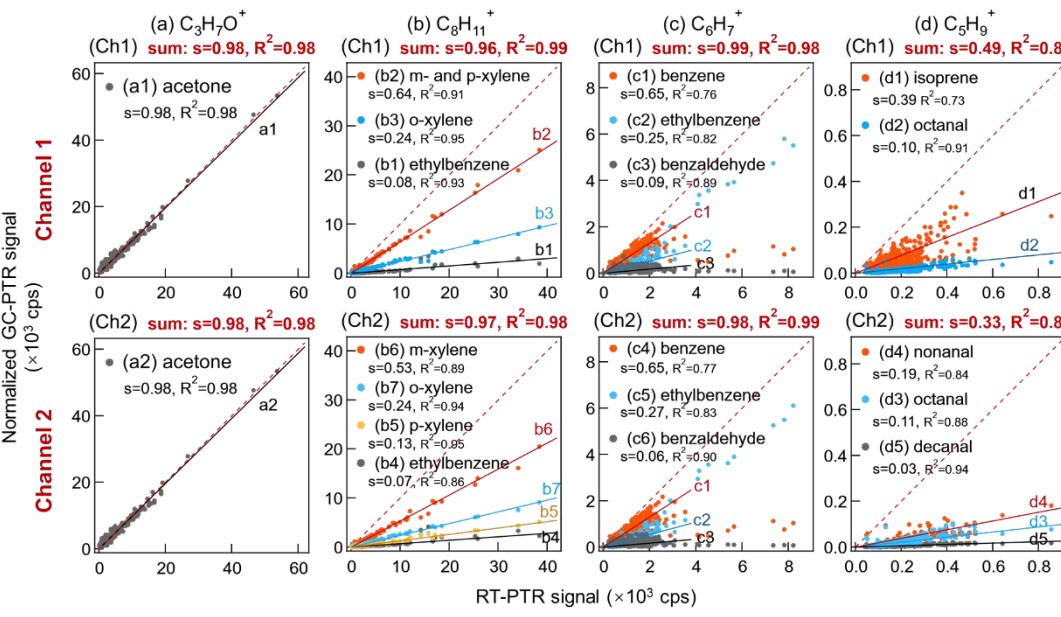



**Figure 4: Inter-comparison of PTR signals between RT-PTR and GC-PTR with a time resolution of one hour. First row: Ch1; Second row: Ch2. (a) $C_3H_7O^+$, acetone in both Ch1 and Ch2. (b) $C_8H_{11}^+$, Ch1: coeluted m-xylene and p-xylene, o-xylene, and ethylbenzene; Ch2: m-xylene, p-xylene, o-xylene, and ethylbenzene. (c) $C_6H_7^+$, ethylbenzene, benzene, and benzaldehyde in both Ch1**
**and Ch2. (d) $C_5H_9^+$, Ch1: isoprene and octanal; Ch2: octanal, nonanal, and decanal. s denotes the slope of the linear fitting and $R^2$ denotes R square. The red dashed line is a 1:1 line for reference.**

Signal comparisons in cps between RT-PTR and GC-PTR measurements of identified VOCs during the entire campaign were performed for both GC channels (Fig. 4). The PTR signals of identified GC-elution peaks were integrated over the elution time for both GC channels to obtain their peak areas (signal counts), and then divided by the sampling time (500 seconds) to obtain
signals (back in cps) as described above in section 2.3, so that they are comparable with the RT-PTR signals, hereinafter referred to as GC-PTR signals.

As shown in Fig. 4a, the slopes of the linear fitting between the PTR signals of directly sampled air and acetone eluted through GC, both at 59.0491 Th, are 0.98 in both Ch1 and Ch2, indicating that atmospheric acetone accounted for ~98% of $C_3H_7O^+$ signals by RT-PTR. The residual ~2% $C_3H_7O^+$ signals by RT-PTR were contributed by propanal, which is normally several
orders of magnitude less abundant in the atmosphere than acetone. Propanal did not show a distinct elution peak in Fig. 3a due to its low abundance in that particular sample but was well detected in samples in other time periods (e.g., Fig. S5). Therefore, the $C_3H_7O^+$ signal in RT-PTR was identified to be acetone and negligible propanal, which is consistent with previous studies (Coggon et al., 2023; Gouw et al., 2003b; Warneke et al., 2003).

Also, in line with previous studies (Coggon et al., 2023; Gouw et al., 2003b; Warneke et al., 2003), the RT-PTR $C_8H_{11}^+$ signals
were dominated by ethylbenzene and xylenes because the sum of ethylbenzene and xylenes explained more than 95% $C_8H_{11}^+$ signals as shown in Fig. 4b. In addition, the $C_8H_{11}^+$ signal was dominated by xylenes, and only ~8% of the total signals was ethylbenzene.

$C_6H_7^+$ in RT-PTR was dominated by benzene, ethylbenzene and benzaldehyde, because the sum of these three VOCs in GC-PTR explained more than 96% $C_6H_7^+$ signals in RT-PTR (Fig. 4c), consistent with earlier observations in Las Vegas (Coggon
et al., 2023). The residual ~4% $C_6H_7^+$ signals were contributed by xylenes, n-propyl benzene and isopropyl benzene, i.e., the small elution peaks labeled as c3-c6 (Ch1) and c10-c13 (Ch2) in Fig. 3c. During most time of our measurements, about 65% of the $C_6H_7^+$ signals by RT-PTR was produced by benzene. However, $C_6H_7^+$ was almost dominated by ethylbenzene when the measured $C_6H_7^+$ signals by RT-PTR were of higher than 4000 cps.

Among VOC species that contributed to $C_5H_9^+$, isoprene eluted only in Ch1, nonanal and decanal only eluted in Ch2, and
octanal eluted in both Ch1 and Ch2. Isoprene only accounted for 39% $C_5H_9^+$ signals by RT-PTR for the entire campaign with a correlation of $R^2$=0.73. Since this was a winter-time urban campaign, it's not surprising that isoprene signal is being swamped by interference here. Even if both GC Ch1 and Ch2 were considered, the sum of isoprene, octanal (average of Ch1 and Ch2), nonanal, and decanal only explained ~72% $C_5H_9^+$ signals. Although many VOCs that could produce $C_5H_9^+$ remain unidentified (Fig. 3d), we can conclude that $C_5H_9^+$ signals in RT-PTR are not suitable for characterizing isoprene concentrations in our
measurement environment.



### 3.3 Classification of RT-PTR ions

As discussed above, among the 176 PTR signals with chromatographic peaks, 22 were not properly characterized by the GC system. We examined the remaining the 154 RT-PTR signals that corresponded to obvious elution peaks in GC-PTR, and attributed VOC identities to each of the 154 m/z values (Table 2). Definitive identifications were achieved for small m/z values,
normally produced by VOCs with high abundances such as aromatic compounds and small OVOCs (C1-C4). However, an unambiguous identification becomes increasingly challenging as the m/z value increases, because the number of isomers increases and the atmospheric abundance decreases in the gas phase as the number of carbon atoms increases. Thus, a number of RT-PTR signal ions especially those that contain more than one O atom and more than five carbon atoms are only attributed with molecular formulas.

According to linear fittings between the RT-PTR signals and the GC-PTR signals of identified VOC(s) for each of the 154 PTR signal ions, an identified VOC or a group of VOCs is arbitrarily considered to dominate the PTR signal, if such a linear fitting results in a slope between 0.9 and 1.1 and $R^2 > 0.9$. To cover as many VOCs that can produce a given PTR signal ion as possible, Table 2 also includes VOCs that generate less than 10% of this PTR signal (noted in "minor"). Ions that were dominated by one specific VOC are grouped into category I; ions that were dominated by one set of VOC isomers are grouped
into category II; and an ion is considered to be category III because of: 1) poor GC elution or non-retention, i.e., the 22 signals that were characterized with a large uncertainty in both GC channels, 2) or the detection of that ion is too complicated (e.g., fragments, water-clusters, and dehydration products) to be used as a quantitative tracer for a compound or family of isomers.

**Table 2** Identity attribution for each RT-PTR signal.

| m/z (Th) | formula | Identity attribution | Classification | Quantification[a] |
|---|---|---|---|---|
| 19.0178 | $H_3O^+$ | reagent ion | reagent ions | / |
| 29.9974 | $NO^+$ | reagent ion | reagent ions | / |
| 31.9893 | $O_2^{+\bullet}$ | reagent ion | reagent ions | / |
| 33.0335 | $CH_5O^+$ | methanol | category I | Avg |
| 37.0284 | $H_5O_2^+$ | reagent ion | reagent ions | / |
| 39.0229 | $C_3H_3^+$ | fragments from dozens of compounds | category III | Ch1 |
| 41.0386 | $C_3H_5^+$ | fragments from dozens of unknown compounds | category III | NC[b] |
| 42.0338 | $C_2H_4N^+$ | acetonitrile | category I | Avg |
| 43.0178 | $C_2H_3O^+$ | acetic acid, ethyl acetate<br>minor: glycolaldehyde, methyl formate, and acetone | category III | Avg |
| 43.0542 | $C_3H_7^+$ | isopropanol<br>minor: acetone and other unknown compounds | category I | Avg |





| | | acetaldehyde | | |
|---|---|---|---|---|
| 45.0335 | $C_2H_5O^+$ | minor: ethanol, 4-methyl-2-pentanone, and other unknown compounds | category I | Ch1 |
| 47.0491 | $C_2H_7O^+$ | ethanol and dimethyl ether<br>minor: dimethyl carbonate | category II | Ch1 |
| 50.0151 | $C_4H_2^+$ | fragments from dozens of compounds | category III | Avg |
| 51.0229 | $C_4H_3^+$ | fragments from dozens of compounds | category III | Avg |
| 51.0441 | $CH_7O_2^+$ | methanol | category I | Avg |
| 53.0022 | $C_3HO^+$ | fragments from dozens of compounds | category III | Ch1 |
| 53.0386 | $C_4H_5^+$ | fragments from dozens of compounds including tetrahydrofuran, butanal, methyl-ethyl-ketone | category III | Avg |
| 54.0338 | $C_3H_4N^+$ | acrylonitrile | category I | Avg |
| 55.0390 | $H_7O_3^+$ | reagent ion | reagent ions | / |
| 55.0542 | $C_4H_7^+$ | fragments from dozens of substances including tetrahydrofuran, butanal, methyl-ethyl-ketone, hexanal, nonanal, decanal and other unknown compounds | category III | Ch2 |
| 56.0495 | $C_3H_6N^+$ | propanenitrile | category I | Avg |
| 57.0335 | $C_3H_5O^+$ | acrolein<br>minor: butanal | category I | Avg |
| 57.0699 | $C_4H_9^+$ | C4-alkene and fragments from hydrocarbons, butyl alcohol, tert-butyl methyl ether, nonanal, decanal and other unknown compounds | category III | Ch1 |
| 59.0491 | $C_3H_7O^+$ | acetone<br>minor: propanal | category I | Avg |
| 60.0444 | $C_2H_6NO^+$ | acetamide and methyl-formamide | category II | Ch2 |
| 61.0284 | $C_2H_5O_2^+$ | acetic acid, ethyl acetate<br>minor: glycolaldehyde, and methyl formate | category III | Avg |
| 62.9632 | $CClO^+$ | methylene chloride ($CH_2Cl_2$) and other unknown compounds | category III | NC |
| 63.0229 | $C_5H_3^+$ | fragments from dozens of compounds | category III | Ch1 |
| 63.0441 | $C_2H_7O_2^+$ | acetaldehyde<br>minor: ethanol | category I | Ch1 |
| 65.0386 | $C_5H_5^+$ | fragments from aromatic compounds | category III | Avg |



| | | | | |
|---|---|---|---|---|
| 65.0597 | $C_2H_9O_2^+$ | ethanol and dimethyl ether<br>minor: dimethyl carbonate | category II | Ch1 |
| 67.0542 | $C_5H_7^+$ | fragments from dozens of compounds including nonanal, decanal, $C_5H_8O$ carbonyls, and other unknown compounds | category III | Ch2 |
| 68.0257 | $C_4H_4O^+$ | furan | category I | Avg |
| 68.0495 | $C_4H_6N^+$ | $C_4H_5N$ nitriles | category II | Avg |
| 69.0335 | $C_4H_5O^+$ | furan<br>minor: $C_4H_6O_2$ and $C_4H_8O_3$ isomers | category I | Avg |
| 69.0699 | $C_5H_9^+$ | isoprene, octanal, nonanal, decanal, $C_5H_{10}O$ carbonyl compounds, and other unknown compounds | category III | NC |
| 70.0651 | $C_4H_8N^+$ | butane nitrile, isobutyronitrile, and unknown $C_4H_7N$ or $C_4H_9NO$ | category III | Avg |
| 71.0491 | $C_4H_7O^+$ | methyl vinyl ketone, tetrahydrofuran, methacrolein, crotonaldehyde | category II | Avg |
| 71.0855 | $C_5H_{11}^+$ | C5-alkenes and fragments from larger compounds | category III | NC |
| 72.0444 | $C_3H_6NO^+$ | acrylonitrile and propanamide | category II | Avg |
| 73.0495 | $H_9O_4^+$ | reagent ion | reagent ions | / |
| 73.0648 | $C_4H_9O^+$ | methyl ethyl ketone<br>minor: butanal, tetrahydrofuran, methyl tert-butyl ether, and unknown compounds | category I | Ch1 |
| 74.0600 | $C_3H_8NO^+$ | propanenitrile and propanamide | category III | Ch2 |
| 75.0441 | $C_3H_7O_2^+$ | acetol<br>minor: propanoic acid, acrolein | category I | Avg |
| 77.0233 | $C_2H_5O_3^+$ | unknown compounds | category III | NC |
| 77.0597 | $C_3H_9O_2^+$ | acetone<br>minor: propanal | category I | Avg |
| 78.0464 | $C_6H_6^+$ | benzene | category I | Avg |
| 79.0390 | $C_2H_7O_3^+$ | acetic acid, ethyl acetate<br>minor: glycolaldehyde, methyl formate, and other unknown compounds | category III | Avg |



| | | | | |
|---|---|---|---|---|
| | | benzene, ethylbenzene, benzaldehyde | | |
| 79.0542 | $C_6H_7^+$ | minor: xylenes, n-propyl benzene and isopropyl benzene | category III | Avg |
| 80.0495 | $C_5H_6N^+$ | pyridine | category I | Avg |
| 81.0335 | $C_5H_5O^+$ | cyclopentadienone | category I | Ch2 |
| 81.0699 | $C_6H_9^+$ | fragments from dozens of substances including monoterpenes, octanal, nonanal, decanal, $C_6H_{10}O$ carbonyls | category III | Ch2 |
| 82.9450 | $CCl_2H^+$ | methylene chloride ($CH_2Cl_2$), trichloromethane ($CHCl_3$), and other unknown compounds | category III | Avg |
| 83.0491 | $C_5H_7O^+$ | $C_5H_6O$ or/and $C_5H_8O_2$ compounds and other unknown compounds | category III | NC |
| 83.0855 | $C_6H_{11}^+$ | C6-alkenes, $C_6H_{12}O$ carbonyl compounds, nonanal, and decanal | category III | Ch2 |
| 84.0808 | $C_5H_{10}N^+$ | C5-nitrile and $C_5H_{11}NO$ compounds | category III | Avg |
| 85.0648 | $C_5H_9O^+$ | $C_5H_{10}O_2$ compounds or/and $C_5H_8O$ carbonyl compounds | category III | Avg |
| 85.1012 | $C_6H_{13}^+$ | C6-alkenes and fragments from larger compounds | category III | Ch1 |
| 86.0600 | $C_4H_8NO^+$ | $C_4H_5N$ nitriles | category II | Avg |
| 87.0441 | $C_4H_7O_2^+$ | $C_4H_6O_2$ and $C_4H_8O_3$ isomers | category III | Avg |
| 87.0804 | $C_5H_{11}O^+$ | $C_5H_{10}O$ carbonyl compounds | category II | Avg |
| 88.0393 | $C_3H_6NO_2^+$ | acetamide | category I | Ch2 |
| 88.0757 | $C_4H_{10}NO^+$ | butane nitrile, isobutyronitrile, and unknown $C_4H_7N$ or $C_4H_9NO$ | category III | Avg |
| 89.0597 | $C_4H_9O_2^+$ | ethyl acetate minor: methyl vinyl ketone, butyric acid | category I | Avg |
| 91.0390 | $C_3H_7O_3^+$ | dimethyl carbonate | category I | Avg |
| 91.0542 | $C_7H_7^+$ | toluene, xylenes, ethylbenzene, ethyl-methyl-benzenes, n-propyl benzene | category III | Avg |
| 91.0754 | $C_4H_{11}O_2^+$ | methyl ethyl ketone; minor: butanal, tetrahydrofuran, methyl tert-butyl ether, and unknown compounds | category I | Ch1 |
| 92.0621 | $C_7H_8^+$ | toluene | category I | Avg |



| | | | | |
|---|---|---|---|---|
| 93.0546 | $C_3H_9O_3^+$ | acetol<br>minor: propanoic acid | category I | Avg |
| 93.0699 | $C_7H_9^+$ | toluene, ethyl-methyl-benzenes<br>minor: monoterpenes | category III | Avg |
| 95.0339 | $C_2H_7O_4^+$ | unknown compounds | category III | NC |
| 95.0491 | $C_6H_7O^+$ | benzene, toluene, ethylbenzene, benzaldehyde, ethyl-methyl-benzenes, phenol | category III | Avg |
| 95.0855 | $C_7H_{11}^+$ | C7-alkenes and $C_7H_{12}O$ carbonyl compounds | category III | Ch2 |
| 97.0284 | $C_5H_5O_2^+$ | $C_5H_4O_2$ or/and $C_5H_6O_3$ compounds | category III | Ch2 |
| 97.0495 | $C_2H_9O_4^+$ | acetic acid, ethyl acetate<br>minor: glycolaldehyde, and methyl formate | category III | Avg |
| 97.0648 | $C_6H_9O^+$ | $C_6H_8O$ or/and $C_6H_{10}O_2$ compounds | category III | Ch2 |
| 97.1012 | $C_7H_{13}^+$ | C7-alkenes, $C_7H_{14}O$ carbonyl compounds, and decanal | category III | Ch2 |
| 99.0077 | $C_4H_3O_3^+$ | maleic anhydride ($C_4H_2O_3$) and $C_6H_6O_2$ isomers and other unknown compounds | category III | NC |
| 99.0804 | $C_6H_{11}O^+$ | $C_6H_{12}O_2$ or/and $C_6H_{10}O$ carbonyl compounds | category III | Avg |
| 99.1168 | $C_7H_{15}^+$ | C7-alkenes and fragments from larger compounds | category III | Ch2 |
| 101.0597 | $C_5H_9O_2^+$ | $C_5H_6O$ or/and $C_5H_8O_2$ compounds and other unknown compounds | category III | NC |
| 101.0961 | $C_6H_{13}O^+$ | $C_6H_{12}O$ carbonyl compounds | category II | Avg |
| 102.0913 | $C_5H_{12}NO^+$ | $C_5H_9N$ and $C_5H_{11}NO$ isomers | category III | Avg |
| 103.0754 | $C_5H_{11}O_2^+$ | $C_5H_{10}O_2$ compounds or/and $C_5H_8O$ carbonyl compounds | category III | Avg |
| 104.0495 | $C_7H_6N^+$ | benzonitrile | category I | Avg |
| 105.0335 | $C_7H_5O^+$ | benzaldehyde and acetophenone | category III | Ch2 |
| 105.0546 | $C_4H_9O_3^+$ | $C_4H_6O_2$ or/and $C_4H_8O_3$ compounds | category III | Ch2 |
| 105.0699 | $C_8H_9^+$ | styrene, ethylbenzene, xylene, ethyl-methyl-benzenes, trimethylbenzenes, isopropyl benzene | category III | Avg |
| 105.0910 | $C_5H_{13}O_2^+$ | $C_5H_{10}O$ carbonyl compounds | category II | Avg |
| 106.0777 | $C_8H_{10}^+$ | ethylbenzene, xylenes | category II | Avg |
| 107.0491 | $C_7H_7O^+$ | benzaldehyde | category I | Avg |
| 107.0703 | $C_4H_{11}O_3^+$ | ethyl acetate | category I | Avg |



| | | | | |
|---|---|---|---|---|
| 107.0855 | $C_8H_{11}^+$ | ethylbenzene, xylenes<br>minor: $C_8H_{12}O$ and $C_8H_{14}O_2$ isomers | category II | Avg |
| 109.0495 | $C_3H_9O_4^+$ | dimethyl carbonate | category I | Avg |
| 109.0648 | $C_7H_9O^+$ | $C_7H_8O$ compounds | category II | Avg |
| 109.1012 | $C_8H_{13}^+$ | C8-alkenes and $C_8H_{14}O$ carbonyl compounds | category III | Ch2 |
| 111.0441 | $C_6H_7O_2^+$ | $C_6H_6O_2$ or/and $C_6H_8O_3$ compounds | category III | Ch1 |
| 111.0804 | $C_7H_{11}O^+$ | $C_7H_{10}O$ or/and $C_7H_{12}O_2$ compounds | category III | Ch2 |
| 111.1168 | $C_8H_{15}^+$ | C8-alkenes and $C_8H_{16}O$ carbonyl compounds | category III | Ch2 |
| 113.0233 | $C_5H_5O_3^+$ | $C_6H_6O_2$ or/and $C_6H_8O_3$ compounds | category III | Ch1 |
| 113.0961 | $C_7H_{13}O^+$ | $C_7H_{14}O_2$ or/and $C_7H_{12}O$ carbonyl compounds | category III | Ch2 |
| 113.1325 | $C_8H_{17}^+$ | C8-alkenes and fragments from larger compounds | category III | Ch2 |
| 115.0390 | $C_5H_7O_3^+$ | $C_5H_4O_2$ isomers | category II | Ch2 |
| 115.0754 | $C_6H_{11}O_2^+$ | $C_6H_8O$ or/and $C_6H_{10}O_2$ compounds | category III | Ch2 |
| 115.1117 | $C_7H_{15}O^+$ | $C_7H_{14}O$ carbonyl compounds | category II | Avg |
| 116.9060 | $CCl_3^+$ | carbon tetrachloride ($CCl_4$) and<br>trichloromonofluoromethane ($CCl_3F$) | category III | Ch1 |
| 117.0910 | $C_6H_{13}O_2^+$ | $C_6H_{12}O_2$ or/and $C_6H_{10}O$ carbonyl compounds | category III | Avg |
| 117.0182 | $C_4H_5O_4^+$ | maleic anhydride ($C_4H_2O_3$) and $C_6H_6O_2$ isomers and<br>other unknown compounds | category III | NC |
| 119.0703 | $C_5H_{11}O_3^+$ | $C_5H_6O$ and $C_5H_8O_2$ isomers and other unknown<br>compounds | category III | NC |
| 119.0855 | $C_9H_{11}^+$ | $C_9H_{10}$ aromatic compounds | category II | Ch2 |
| 119.1067 | $C_6H_{15}O_2^+$ | $C_6H_{12}O$ carbonyl compounds | category II | Avg |
| 120.0934 | $C_9H_{12}^+$ | trimethylbenzenes, ethyl-methyl-benzenes, isopropyl<br>benzene, n-propyl benzene | category II | Avg |
| 121.0648 | $C_8H_9O^+$ | acetophenone<br>minor: methyl-benzaldehydes | category I | Ch2 |
| 121.1012 | $C_9H_{13}^+$ | trimethylbenzenes, ethyl-methyl-benzenes, isopropyl<br>benzene, n-propyl benzene | category II | Avg |
| 122.0600 | $C_7H_8NO^+$ | benzonitrile | category I | Avg |
| 123.0441 | $C_7H_7O_2^+$ | unknown compounds | category III | NC |
| 123.0652 | $C_4H_{11}O_4^+$ | unknown compounds | category III | NC |
| 123.0804 | $C_8H_{11}O^+$ | $C_8H_{10}O$ aromatic isomers | category II | Ch2 |



| 123.1168 | $C_9H_{15}^+$ | C9-alkenes and $C_9H_{16}O$ carbonyl compounds | category III | Ch2 |
|---|---|---|---|---|
| 125.0597 | $C_7H_9O_2^+$ | benzaldehyde | category I | Avg |
| 125.0961 | $C_8H_{13}O^+$ | $C_8H_{12}O$ or/and $C_8H_{14}O_2$ compounds | category III | Ch2 |
| 125.1325 | $C_9H_{17}^+$ | C9-alkenes and $C_9H_{18}O$ carbonyl compounds | category III | Ch2 |
| 127.1117 | $C_8H_{15}O^+$ | $C_8H_{16}O_2$ or/and $C_8H_{14}O$ carbonyl compounds | category III | Ch2 |
| 127.1481 | $C_9H_{19}^+$ | C9-alkenes and fragments from larger compounds | category III | Ch2 |
| 129.0546 | $C_6H_9O_3^+$ | $C_6H_6O_2$ or/and $C_6H_8O_3$ compounds | category III | Ch1 |
| 129.0699 | $C_{10}H_9^+$ | naphthalene | category I | Ch2 |
| 129.0910 | $C_7H_{13}O_2^+$ | $C_7H_{10}O$ or/and $C_7H_{12}O_2$ compounds | category III | Ch2 |
| 129.1274 | $C_8H_{17}O^+$ | $C_8H_{16}O$ carbonyl compounds | category II | Avg |
| 131.0339 | $C_5H_7O_4^+$ | $C_6H_6O_2$ or/and $C_6H_8O_3$ compounds | category III | Ch1 |
| 131.1067 | $C_7H_{15}O_2^+$ | $C_7H_{14}O_2$ or/and $C_7H_{12}O$ carbonyl compounds | category III | Ch2 |
| 133.0859 | $C_6H_{13}O_3^+$ | $C_6H_8O$ and $C_6H_{10}O_2$ isomers | category III | Ch2 |
| 133.1012 | $C_{10}H_{13}^+$ | $C_{10}H_{12}$ aromatic compounds | category II | Ch2 |
| 133.1223 | $C_7H_{17}O_2^+$ | $C_7H_{14}O$ carbonyl compounds | category II | Avg |
| 135.0804 | $C_9H_{11}O^+$ | $C_9H_{11}O$ isomers | category II | Avg |
| 135.1016 | $C_6H_{15}O_3^+$ | unknown compounds | category III | NC |
| 135.1168 | $C_{10}H_{15}^+$ | $C_{10}H_{14}$ aromatic compounds<br>minor: $C_{10}H_{16}O$ or/and $C_{10}H_{18}O_2$ compounds | category II | Avg |
| 136.0757 | $C_8H_{10}NO^+$ | $C_8H_9NO$ isomers | category II | Avg |
| 137.1325 | $C_{10}H_{17}^+$ | monoterpenes<br>minor: $C_{10}H_{19}O$ aldehydes and ketones, and hydrocarbons | category II | Ch2 |
| 139.0754 | $C_8H_{11}O_2^+$ | acetophenone<br>minor: methyl-benzaldehydes | category I | Avg |
| 139.1117 | $C_9H_{15}O^+$ | $C_9H_{14}O$ or/and $C_9H_{16}O_2$ compounds | category III | Ch2 |
| 139.1481 | $C_{10}H_{19}^+$ | C10-alkenes and $C_{10}H_{20}O$ carbonyl compounds | category III | Ch2 |
| 141.0546 | $C_7H_9O_3^+$ | unknown compounds | category III | NC |
| 141.0910 | $C_8H_{13}O_2^+$ | unknown compounds | category III | NC |
| 141.1274 | $C_9H_{17}O^+$ | $C_9H_{18}O_2$ or/and $C_9H_{16}O$ carbonyl compounds | category III | Ch2 |
| 141.1638 | $C_{10}H_{21}^+$ | C10-alkenes and fragments from larger compounds | category III | Ch2 |
| 143.0855 | $C_{11}H_{11}^+$ | 1-methyl-naphthalene and other unknown compounds | category III | NC |
| 143.1067 | $C_8H_{15}O_2^+$ | $C_8H_{12}O$ or/and $C_8H_{14}O_2$ compounds | category III | Ch2 |



| 143.1430 | $C_9H_{19}O^+$ | $C_9H_{18}O$ carbonyl compounds | category II | Ch2 |
|---|---|---|---|---|
| 145.1223 | $C_8H_{17}O_2^+$ | $C_8H_{16}O_2$ or/and $C_8H_{14}O$ carbonyl compounds | category III | Ch2 |
| 145.9685 | $C_6Cl_2H_4^+$ | dichlorobenzene | category I | Avg |
| 146.9763 | $C_6Cl_2H_5^+$ | dichlorobenzene | category I | Avg |
| 147.1168 | $C_{11}H_{15}^+$ | $C_{11}H_{14}$ aromatic compounds | category II | Ch2 |
| 147.1380 | $C_8H_{19}O_2^+$ | $C_8H_{16}O$ carbonyl compounds | category II | Avg |
| 149.1325 | $C_{11}H_{17}^+$ | aromatic $C_{11}H_{16}$ isomers<br>minor: $C_{11}H_{18}O$ or/and $C_{11}H_{20}O_2$ compounds | category II | Ch2 |
| 153.0910 | $C_9H_{13}O_2^+$ | unknown compounds | category III | NC |
| 153.1274 | $C_{10}H_{17}O^+$ | $C_{10}H_{16}O$ or/and $C_{10}H_{18}O_2$ compounds and other unknown compounds | category III | NC |
| 153.1638 | $C_{11}H_{21}^+$ | C11-alkenes and $C_{11}H_{22}O$ carbonyl compounds | category III | Ch2 |
| 155.1430 | $C_{10}H_{19}O^+$ | $C_{10}H_{18}O$ aldehydes and ketones | category II | Ch2 |
| 155.1794 | $C_{11}H_{23}^+$ | C11-alkenes and fragments from larger compounds | category III | Ch2 |
| 157.1223 | $C_9H_{17}O_2^+$ | $C_9H_{14}O$ or/and $C_9H_{16}O_2$ compounds | category III | Ch2 |
| 157.1587 | $C_{10}H_{21}O^+$ | $C_{10}H_{20}O$ aldehydes and ketones | category II | Ch2 |
| 159.1380 | $C_9H_{19}O_2^+$ | $C_9H_{18}O_2$ or/and $C_9H_{16}O$ carbonyl compounds | category III | Ch2 |
| 161.1325 | $C_{12}H_{17}^+$ | $C_{12}H_{16}$ aromatic compounds | category II | Ch2 |
| 161.1536 | $C_9H_{21}O_2^+$ | $C_9H_{18}O$ carbonyl compounds | category II | Ch2 |
| 163.1481 | $C_{12}H_{19}^+$ | aromatic $C_{12}H_{18}$ isomers | category II | Ch2 |
| 165.1638 | $C_{12}H_{21}^+$ | C12-alkenes or/and larger carbonyl compounds | category III | Ch2 |
| 167.0550 | $C_5H_{11}O_6^+$ | unknown compounds | category III | NC |
| 171.1380 | $C_{10}H_{19}O_2^+$ | $C_{10}H_{16}O$ or/and $C_{10}H_{18}O_2$ compounds and other unknown compounds | category III | NC |
| 171.1743 | $C_{11}H_{23}O^+$ | $C_{11}H_{22}O$ carbonyl compounds | category II | Ch2 |
| 173.0808 | $C_8H_{13}O_4^+$ | unknown compounds | category III | NC |
| 175.1693 | $C_{10}H_{23}O_2^+$ | $C_{10}H_{20}O$ carbonyl compounds | category II | Ch2 |
| 189.1849 | $C_{11}H_{25}O_2^+$ | $C_{11}H_{22}O$ carbonyl compounds | category II | Ch2 |
| 223.0636 | $C_6H_{19}O_3Si_3^+$ | hexamethylcyclotrisiloxane (D₃) | category I | Avg |
| 225.0429 | $C_5H_{17}O_4Si_3^+$ | hexamethylcyclotrisiloxane (D₃) | category I | Avg |
| 241.0742 | $C_6H_{21}O_4Si_3^+$ | hexamethylcyclotrisiloxane (D₃) | category I | Avg |
| 297.0824 | $C_8H_{25}O_4Si_4^+$ | octamethylcyclotetrasiloxane (D₄) | category I | Avg |
| 299.0617 | $C_7H_{23}O_5Si_4^+$ | octamethylcyclotetrasiloxane (D₄) | category I | Avg |



| 301.0410 | $C_6H_{21}O_6Si_4^+$ | octamethylcyclotetrasiloxane (D$_4$) | category I | Avg |
| 315.0930 | $C_8H_{27}O_5Si_4^+$ | octamethylcyclotetrasiloxane (D$_4$) | category I | Avg |
| 355.0700 | $C_9H_{27}O_5Si_5^+$ | decamethylcyclopentasiloxane (D$_5$) | category I | Ch2 |
| 371.1012 | $C_{10}H_{31}O_5Si_5^+$ | decamethylcyclopentasiloxane (D$_5$) | category I | Ch2 |
| 373.0805 | $C_9H_{29}O_6Si_5^+$ | decamethylcyclopentasiloxane (D$_5$) | category I | Ch2 |

Note:

[a] quantification is based on the usage of GC-PTR values of either Ch1, or Ch2, or the average of Ch1 and Ch2 (Avg). 61 signals were quantified using Ch2 because of a relative difference of larger than 10% between GC-Ch1-PTR and RT-PTR; 15 signals were quantified using Ch1 because of a relative difference of larger than 10% between GC-Ch2-PTR and RT-PTR; 78 signals were quantified using the average GC-PTR value of Ch1 and Ch2 because of a relative difference between -10% to 10% in both channels.

[b] "NC" stands for 22 signals that were not properly characterized by either GC channels.

In the following discussion, the quantification of GC-PTR and RT-PTR measurements was achieved by using authentic standards.

Category I contains 45 ions that were dominantly produced by 25 VOC species, because a number of VOC species produced
more than one category I ion. For example, $C_3H_7O^+$ and $C_3H_9O_2^+$ are representative category I ions that can be attributed to be MH$^+$ and [MH+H$_2$O]$^+$ from various reaction channels of acetone in the IMR. The quantification of VOCs according to category I ions in our measurement is deemed to be reliable. As shown in Fig. 5a, taking acetone for instance, the acetone concentrations between RT-PTR and GC-PTR measurements resulted in an excellent linear relationship with a slope of 1.02 and a $R^2$ of 0.95. Category II contains 39 signal ions, each of which was dominantly produced by a group of isomers. $C_8H_{11}^+$ and $C_8H_{10}^+$ are
representative category II ions that are both generated by ethylbenzene and xylenes. Since category II ions are conventionally quantified with the calibration factor of one of the isomers, caution must be taken because isomers undergo proton transfer reactions with different rates ($k_{PTR}$) and subsequent fragmentation patterns in the PTR. Taking C8 aromatics (ethylbenzene and xylenes) for instance, the average calibration factor using $C_8H_{11}^+$ measured from o/m/p-xylene is ~3.3±0.02 (mean ± standard deviation) times of that from ethylbenzene, because $C_8H_{11}^+$ represents ~81.2% of the total signals of all product ions from
xylenes whereas only ~24.7% exists in the case of ethylbenzene in PTR measurements. Adopting the average calibration factor of xylenes (Fig. 5b) resulted in an underestimation of the total concentrations of isomers especially when the ratios of xylene/ethylbenzene were low, whereas adopting the calibration factor of ethylbenzene (Fig. 5c) resulted in a significant overestimation.





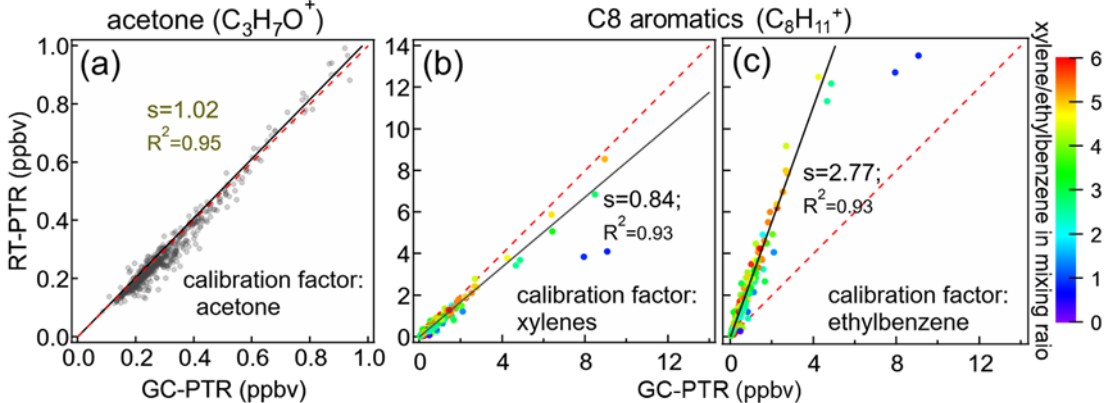

**Figure 5: Mixing ratios of acetone and C8 aromatics (xylenes and ethylbenzene) measured by GC-PTR *v.s.* RT-PTR. The quantification of GC-PTR measurements was achieved by using authentic acetone, xylenes, and ethylbenzene. Also shown are quantification of acetone in the RT-PTR measurement using the calibration factor of $C_3H_7O^+$ derived from authentic acetone (a), and quantification of C8 aromatics in RT-PTR using the calibration factor of $C_8H_{11}^+$ derived from authentic (b) xylenes and (c) ethylbenzene, respectively. The slope and R square are noted as s and $R^2$, respectively. The red dash line denotes a 1:1 line for reference.**

Including the 22 ions that were not well characterized by the GC system, category III contains 92 PTR ions that were produced by various non-isomeric VOCs. Typical examples are $C_6H_7^+$ and $C_5H_9^+$ that are traditionally used for benzene and isoprene quantification, respectively. Up-limits for Category III ions were normally obtained since there could be contributors without assigned identities.

### 3.4 Quantification of selected VOCs using either non-MH⁺ or non-Category I ions

Our discussion in the previous section suggests that only a limited number of MH⁺ ions in RT-PTR can be used to reliably derive atmospheric concentrations of a VOC species (M). Clearly, it is also impractical to couple every single PTR-MS with a GC for a better quantification. Nevertheless, the overall product ion distributions of various reaction channels for an atmospheric species are expected to vary only slightly under a given PTR-MS setting (Jensen et al., 2023), especially during one campaign. Indeed, the signal ion distributions obtained in this study are overall consistent with those obtained by Jensen et al. (2023) under an E/N of 160 Td, but show higher water-clustering products and lower fragments and de-watering products. Here we propose additional PTR-MS calibration steps with authentic VOC standards, together with the understanding obtained in this study with the help of gas-chromatographic pre-separation, to derive more reliable concentrations for a number of VOC species solely from PTR-MS measurements.

### 3.4.1 Quantification of benzene and toluene using $C_6H_6^+$ and $C_7H_8^+$, respectively

As discussed above, about 65% of the $C_6H_7^+$ signals by RT-PTR were produced by benzene during most of our measurement time, leading to an unreliable PTR quantification of benzene through $C_6H_7^+$. As proposed by Coggon et al. (2023), we instead quantified benzene using the charge transfer product ion, $C_6H_6^+$ (category I ion), which has not been observed to be produced





from other VOCs so far, rather than the normally used $C_6H_7^+$ (category III ion). The sensitivity for our RT-PTR to benzene is ~3800 cps/ppbv when using $C_6H_7^+$, and is ~840 cps/ppbv when using $C_6H_6^+$. The ratio of $C_6H_7^+$ to $C_6H_6^+$ that we observed for authentic benzene is comparable to Coggon et al. (2023) and Link et al. (2024a). As shown in Fig. 6, the mixing ratios of benzene measured by GC-PTR are used for reference, resulting in a satisfactory linear relationship with a slope of 1.02 and a $R^2$ of 0.98. The severe overestimation of benzene on January 25th and February 24th (Fig. 6, brown line) quantified by the

$C_6H_7^+$ (MH$^+$) signal was due to the high concentrations of ethylbenzene (see Fig. 8).

The quantification of toluene by $C_7H_9^+$ resulted in a slight overestimation of 19% due to the fragmentation of ethyl-methyl-benzenes as shown in Fig. S6.  Using a similar approach as for benzene, the toluene charge transfer product ion $C_7H_8^+$ is more reliable because the slope and $R^2$ of the linear fitting was 0.96 and 0.98, respectively.

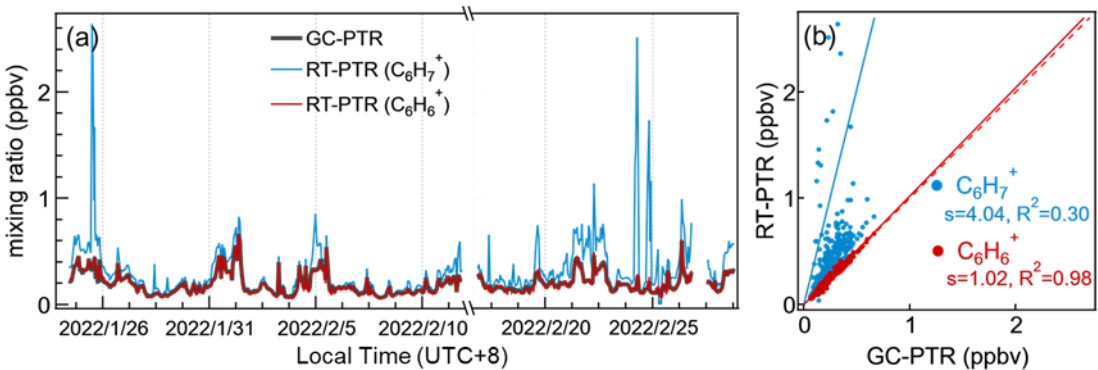


**Figure 6: Inter-comparison of mixing ratios of benzene between GC-PTR measurements and RT-PTR measurements quantified by $C_6H_6^+$ signal and $C_6H_7^+$. The red dashed line denotes a 1:1 line for reference. The slope and R square are noted as s and $R^2$, respectively.**

### 3.4.2 Quantification of aromatic isomers

A matrix (Fig. 7) between common aromatic compounds, a relatively independent group of compounds, and all of their PTR-MS ions was prepared for the sample collected from 16:26:46 to 16:35:07 on 19 February 2022 to investigate the mutual interference between these aromatics, and to seek quantitative correction recommendations based solely on the RT-PTR signals and the distributions of aromatics' product ions. The aromatic compounds discussed here include benzene, phenol, toluene, benzaldehyde, styrene, o/m/p-xylenes, ethylbenzene, acetophenone, trimethylbenzenes, ethyl-methyl-benzenes, n-propyl-

benzene, and iso-propyl-benzenes. Isomers with the same functional groups such as o/m/p-xylenes show almost identical product ion distributions in PTR-MS and are hence considered together. These aromatic VOCs involve 17 product ions. There was an interference on the $C_7H_9^+$ ion due to the fragmentation of monoterpenes ($C_{10}H_{16}$) (Table 2). However, toluene and ethyl-methyl-benzenes explained 96% of the $C_7H_9^+$ RT-PTR signals in our one-month measurement, and monoterpene concentrations were low enough so that they did not represent a significant interference and thus not further considered within

the matrix.





In this matrix, seven ions belonging to category I were not interfered by other substances: $C_6H_6^+$ for benzene and $C_7H_8^+$ for toluene as discussed previously, $C_7H_7O^+$ and $C_7H_9O_2^+$ for benzaldehyde, $C_8H_8^+$ for styrene, and $C_8H_9O^+$ and $C_8H_{11}O_2^+$ for acetophenone. Thus, benzene, toluene, benzaldehyde, styrene, and acetophenone can be accurately quantified using their corresponding category I ions directly. $C_8H_{10}^+$ and $C_8H_{11}^+$, $C_9H_{12}^+$ and $C_9H_{13}^+$ are category II ions, representing the sum of the

C8 and C9 aromatic isomers, respectively. The other six ions belong to category III, among which $C_6H_7O^+$ and $C_8H_9^+$ led to significant and uncorrectable overestimations of phenol and styrene, respectively; and $C_6H_7^+$ and $C_7H_9^+$ led to overestimations of benzene and toluene, respectively.

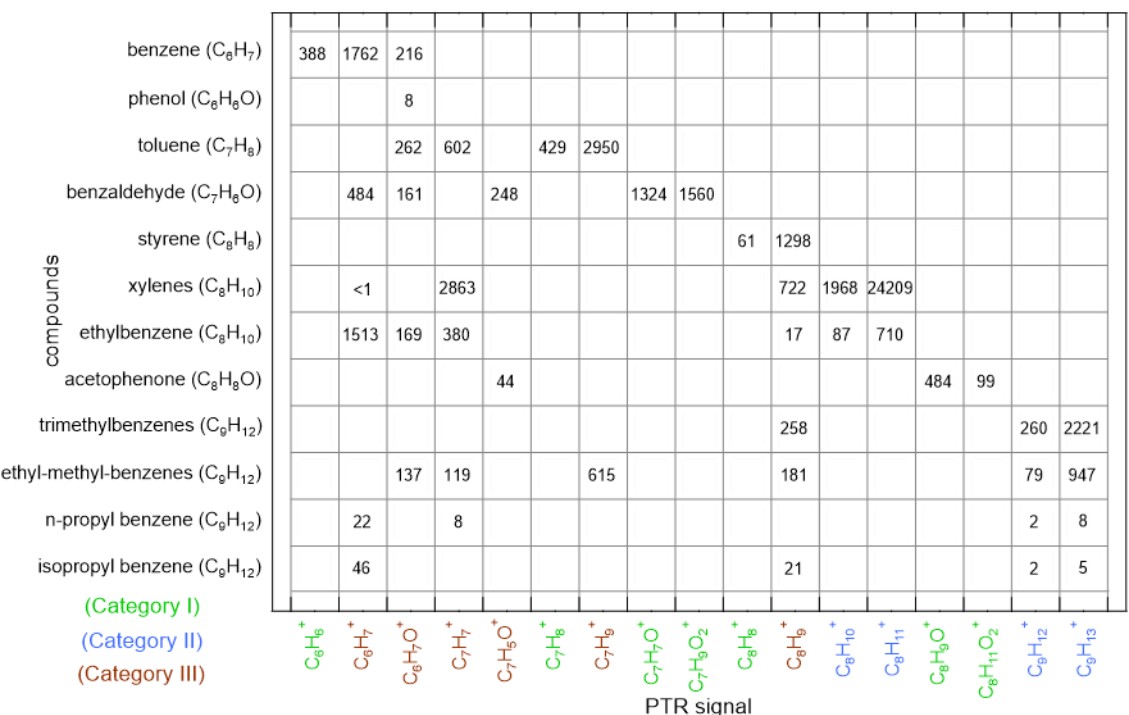

**Figure 7: A representative matrix between aromatic species and their 17 PTR-MS signals (in cps) for a sample collected from 16:26:46 to 16:35:07 on 19 February 2022.**

Allocating the $C_8H_{11}^+$ signal in RT-PTR to xylenes and ethylbenzene relies on the ratio of the charge transfer product $M^+$ to the protonated $MH^+$, which is,

$$r_1 * S[C_8H_{11}^+\_xylenes] + r_2 * S[C_8H_{11}^+\_ethylbenzene] = S[C_8H_{10}^+] \quad (Eq.2)$$

$$S[C_8H_{11}^+\_xylenes] + S[C_8H_{11}^+\_ethylbenzene] = S[C_8H_{11}^+] \quad (Eq.3)$$

where $S[C_8H_{11}^+\_xylenes]$ and $S[C_8H_{11}^+\_ethylbenzene]$ are estimated $C_8H_{11}^+$ signals that are produced from xylenes and ethylbenzene in the RT-PTR, respectively; $r_1$ and $r_2$ are the ratios of $C_8H_{10}^+/C_8H_{11}^+$ produced by authentic xylenes and ethylbenzene, respectively, being 0.0813 and 0.123 under our PTR setting; $S[C_8H_{10}^+]$ and $S[C_8H_{11}^+]$ are signals of $C_8H_{10}^+$ and $C_8H_{11}^+$ in the RT-PTR measurement. The calculated $S[C_8H_{11}^+\_xylenes]$ and $S[C_8H_{11}^+\_ethylbenzene]$ are shown in Fig. 8a and





8b, with comparisons with those measured by GC-PTR. The estimated mixing ratios of xylenes and ethylbenzene were calculated by the calibration factor of xylenes and ethylbenzene, respectively, and are presented in Fig. 8c-8f. The estimated xylene mixing ratios are slightly higher than, i.e., 1.06 times of, the measured values from GC-PTR, and the estimated values of ethylbenzene are slightly lower than, i.e., 0.95 times of the measured ones.

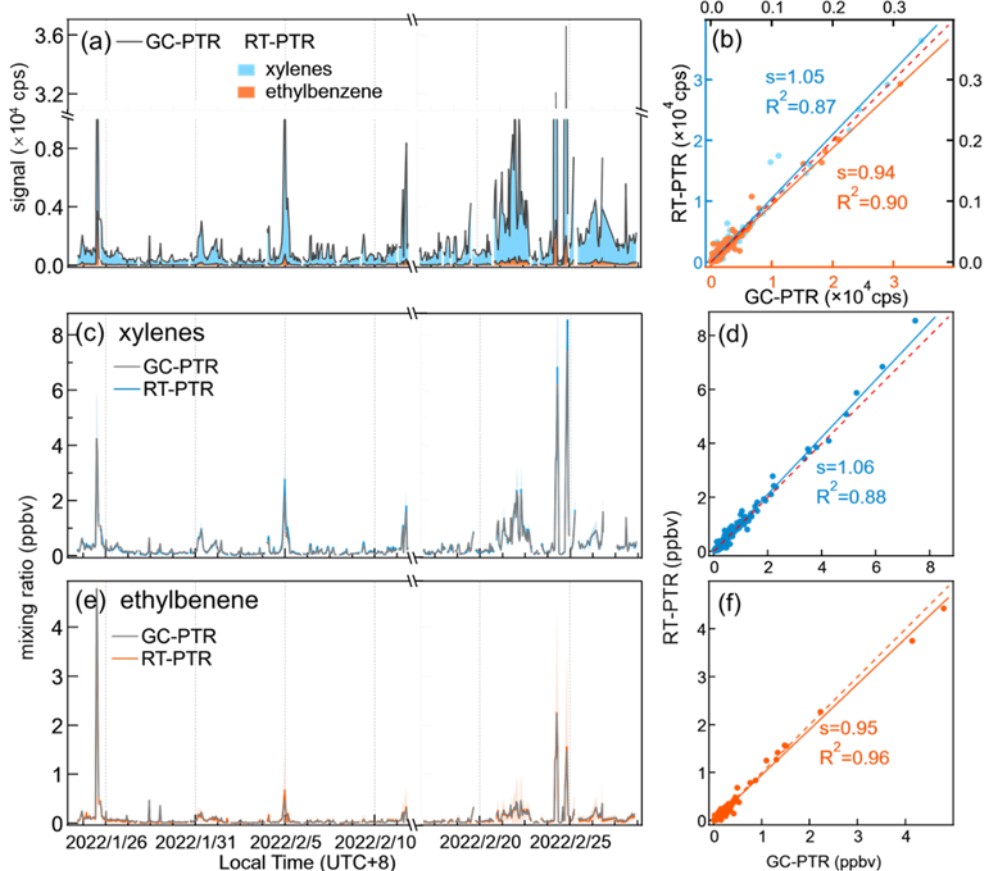


**Figure 8: (a and b) Allocation of $C_8H_{11}^+$ PTR signals to xylenes and ethylbenzene, and mixing ratios of xylenes (c and d) and ethylbenzene (e and f) quantified by the calculated PTR signals and calibration factors derived from authentic compounds. The red dashed line denotes a 1:1 line for reference.**

The extrapolation of $S[C_8H_{11}^+\_ethylbenzene]$, i.e., the $C_8H_{11}^+$ signal that was produced by ethylbenzene in RT-PTR, provides

an opportunity to correct the $C_6H_7^+$ signal (category III) for quantification of benzene by deducting the $C_6H_7^+$ signals generated by interferents (benzaldehyde and ethylbenzene) as follows:

$$S[C_6H_7^+]_{corr} = S[C_6H_7^+] - S[C_7H_7O^+] * r_3 - S[C_8H_{11}^+\_ethylbenzene] * r_4 \quad (Eq.4)$$

where $S[C_6H_7^+]_{corr}$ is corrected $C_6H_7^+$ signals that were produced from benzene in the RT-PTR measurement; $r_3$ and $r_4$ are the ratio of $C_6H_7^+/C_7H_7O^+$ produced by authentic benzaldehyde (0.366) and the ratio of $C_6H_7^+/C_8H_{11}^+$ produced by authentic





ethylbenzene (2.130), respectively; $S[C_6H_7^+]$ and $S[C_7H_7O^+]$ are signals of $C_6H_7^+$ and $C_7H_7O^+$ in the RT-PTR measurement, respectively; $S[C_8H_{11}^+\_ethylbenzene]$ is estimated $C_8H_{11}^+$ that was produced from ethylbenzene as discussed above. The corrected mixing ratios of benzene are shown in Fig. S7. The benzene concentration calculated using the corrected RT-PTR $C_6H_7^+$ signal is characterized with an overestimation of 23% compared to that measured by GC-PTR, potentially due the uncertainties introduced during the multi-step calculation.

Nevertheless, this matrix will change with the product ion distributions (i.e., setting of the PTR-MS) and ambient abundances of various aromatics. Caution must be taken and on-site measurements of ion ratios should be performed when applied to other measurements.

### 3.4.3 Uncorrectable overestimation of isoprene using $C_5H_9^+$ in the urban atmosphere

$C_5H_9^+$, a category III ion that is traditionally used for isoprene quantification by PTR, was suggested to originate from methylbutanal, pentanal, octanal, nonanal and 1-nonene in addition to isoprene in previous studies (Coggon et al., 2023; Vermeuel et al., 2023). However, the GC-PTR chromatogram of $C_5H_9^+$ obtained in Shanghai during winter, 2022 with weakened biogenic sources for isoprene as expected is much more complex (Fig. 3d). As a result, quantifying isoprene in RT-PTR by $C_5H_9^+$ using a PTR-MS calibration factor of isoprene led to an average concentration that is 1.56-fold larger than that measured by GC-PTR (Fig. 9). Since deducting the $C_5H_9^+$ signal generated by octanal, nonanal, and decanal demonstrate an improved accuracy of the isoprene measurement in the forest area (Vermeuel et al., 2023), we make an attempt according to the following formula:

$$S[C_5H_9^+]_{corr} = S[C_5H_9^+] - S[C_8H_{17}O^+] * r_5 - S[C_9H_{19}O^+] * r_6 - S[C_{10}H_{21}O^+] * r_7 \quad (Eq.4)$$

where $S[C_5H_9^+]_{corr}$ is corrected $C_5H_9^+$ signals; $r_5$, $r_6$ and $r_7$ are the ratio of $C_5H_9^+/C_8H_{17}O^+$ produced by octanal (2.961), the ratio of $C_5H_9^+/C_9H_{19}O^+$ produced by nonanal (2.161), and the ratio of $C_5H_9^+/C_{10}H_{21}O^+$ produced by decanal (0.260), respectively; $S[C_5H_9^+]$, $S[C_8H_{17}O^+]$, $S[C_9H_{19}O^+]$, $S[C_{10}H_{21}O^+]$ are signals of $C_5H_9^+$, $C_8H_{17}O^+$, $C_9H_{19}O^+$ and $C_{10}H_{21}O^+$ in the RT-PTR measurement, respectively.

As shown in Fig. 9, there is still a gap between the isoprene concentration calculated by the corrected $C_5H_9^+$ signal in RT-PTR and the concentration measured by GC-PTR, indicating that considering identified interferences of octanal, nonanal, and decanal is not sufficient for isoprene correction in RT-PTR detection in our measurement in urban Shanghai.

Another approach of $C_5H_9^+$ signal correction was tested, which assumes that the isoprene concentration is zero at nighttime so that the $C_5H_9^+$ signal at night is generated entirely by interferences, and the extent of interference is proportional to the sum of the m/z 125 and 111 signals generated from aldehydes, i.e., the dehydrated signal of $C_8H_{16}OH^+$ and $C_9H_{18}OH^+$, respectively (Coggon et al., 2023) . Our corrected $C_5H_9^+$ signal had a large number of negative values (Fig. S8a), probably resulting from the abundant isoprene at night emitted from anthropogenic activities that was verified by GC measurement as shown in Fig. S8b.



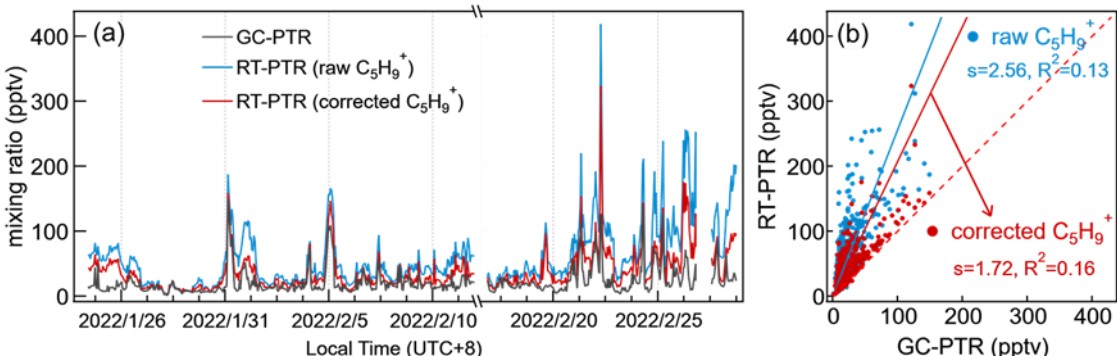

**Figure 9: Inter-comparison of mixing ratios of isoprene between GC-PTR measurements and RT-PTR measurements quantified by raw and corrected $C_5H_9^+$ signals. The red dashed line denotes a 1:1 line for reference. The slope and R square are noted as s and $R^2$, respectively.**

## 4 Conclusion

PTR-MS enables real-time VOC measurements with a high time resolution, but its inherent drawbacks include the inability to distinguish isomers and the non-exclusivity between $MH^+$ signals and concentrations of a VOC species (M). Signals such as $[MH-C_xH_y)]^+$, $[MH-(H_2O)]^+$, $[MH+n(H_2O)]^+$, and $M^+$ complicate the interpretation of the PTR mass spectrum and cause quantification bias.

In this study, we sampled and preseparated ambient VOC molecules via chromatographic techniques, prior to PTR measurements, to gain insight into how a single ion measured by the PTR is produced by multiple VOC species. We provided a widely applicable reference table for attributing the PTR signal to contributing VOC species with as many PTR signals and VOCs as possible. The PTR signals are grouped into three categories according to the complexity of their potential identities. 45 decent signal ions (category I) were generated from only one VOC species, and can be used for a reliable quantification; 39 signal ions (category II) were produced from a group of isomers, and can be used for quantifying the sum of isomers with an inevitable uncertainty if a calibration factor for one specific isomer is used; 92 signal ions (category III) were yielded from more than one non-isomeric species and thus the signal of a category III ion merely gives an upper limit of a VOC concentration. PTR-MS is widely applied to simultaneously measure hundreds of VOCs, and inaccurate quantifications of VOCs may mislead source apportionments derived from Positive Matrix Factorization analysis (Vlasenko et al., 2009), skew ozone formation sensitivity by the EKMA curve (Huang et al., 2024; Li et al., 2024b), and misguide estimation of atmospheric oxidation capacity based on VOC concentrations (Wang et al., 2022). For example, the overestimation of isoprene, especially in urban areas, will cause significant errors in the calculation of its flux and global budget (Eerdekens et al., 2009; Kalogridis et al., 2014). Since our recommended correction depends on the specific measurement time and location and the instrument setting, it is therefore necessary to carry out more measurements under various atmospheric environments such as industrial estates and rural areas. In addition, there is a need to measure at different PTR settings to better understand how signal distributions vary for different VOCs.



*Supplement*. The supplement related to this article is available online at: XXX

*Author contributions*. LW designed the study. MSC, BML and DW provided support for the experimental setup. YWW, CL, YYL and SJY conducted the filed measurement. YZ analyzed the data and wrote the draft, and LW revised the paper with contributions from all co-authors.

*Competing interest*. The authors declare that they have no conflicts of interest.

*Financial support*. This research was supported by the National Natural Science Foundation of China (21925601).

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
