# Peer review of "Interpretation of mass spectra by a Vocus proton transfer reaction mass spectrometer (PTR-MS) at an urban site: insights from gas-chromatographic pre-separation"

_EGUsphere, 2025_

## Author Response (AR1)

We thank the Referee for his/her time and constructive feedback on the manuscript. We believe these comments have helped to improve the organization and presentation of our manuscript. Below we show the reviewer's comments in black and our responses in blue.

**Response to Reviewer 1**

The present manuscript by Zhang et al. entitled "Interpretation of mass spectra by a Vocus proton transfer reaction mass spectrometer (PTR-MS) at an urban site: insights from gaschromatic pre-separation" describes a study where urban air in Shanghai is analyzed using a combination of GC with PTR-MS. Although clearly and well written this manuscript reads like a chapter of a master thesis, that is, it is much too long and much too detailed (in particular the lengthy introduction, the detailed description of standard measuring set ups and procedures and the detailed presentation of results, Table2, without a discussion about general implications concerning the relevance in terms of for instance atmospheric pollution, too many references) in relation to the novelty and originality of the results. The upgrading of a PTR-MS instrument (or other linear mass spectrometers) by coupling it with a standard GC has been carried out and described already several times in literature and thus is nothing new and nowadays a standard procedure when trying to analyse composite samples, in particular when using commercial off the shelf instruments. Moreover, this standard procedure has been applied in the present case to a very specific sample, that is the air measurements were carried out on the rooftop at the Jiangwan campus (Fudan University) from Jannuary 24 to February 28 in 2022. Thus results obtained and presented will only apply to this situation of atmospheric composition and experiemntal setup. The authors themselves argue along this line "Since our recommended correction depends on the specific measurement time and location and the instrument setting,it is therefore necessary to carry out more measurements under various atmospheric environments such as industrial estates and rural areas." So although quite elaborate this experimental study is of rather limited general use and general interest to readers interested in the geoscience community. Nevertheless, it contains a careful study concerning details regarding the PTR MS method and thus publication is possible but the manuscript should be shortened drastically (some of the material could be made available via supplements).

We thank this reviewer for his/her constructive comments. We have revised our manuscript by

1. shortening the introduction,
2. moving Table 1 in the main text to supplementary Table S1 in the revised manuscript,
3. and moving the detailed description of measuring set ups and procedures to the revised supplement.

**Response to Reviewer 2**

Zhang et al. reported a very comprehensive characterization on the measurement capability of volatile organic compounds (VOCs) by a Vocus proton-transfer-reaction mass spectrometer (PTR-MS) with and without gas-chromatographic pre-separation. Field data were used for such a characterization, with a well-designed one-hour protocol to switch between real-time PTR (RT-PTR), gas-chromatographic PTR (GC-PTR), and GC electron-impact ionization time-of-light MS (GC-EI-ToF) measurements. Product identification and quantification were performed with rigorous data analysis routines. The authors segregated hundreds of measured VOCs with three

categories (I, II, and III) based on confidence of identification/quantification. In addition to some known interferences in the literature being further confirmed, the authors also revealed some new insights about potential caveats with the RT-PTR approach to measure ambient VOCs, and provided solutions when possible. Focusing on aromatic species, the authors thoroughly evaluated the performance of both RT-PTR and GC-PTR in ambient measurements, and proposed that using ions other than the protonated ions might be better for the quantification of some aromatic species. The study was well designed and executed, and the manuscript was also well written. I see it more or less publishable as is without too many major comments, but do provide below a few for the authors to consider elaborating for the interest of some readers.

We thank this reviewer for his/her positive and constructive comments.

**Comment 1**: The authors stated that the gray data points in Figure 2a were species that did not elute in either channel of the GC, and they might be of high carbon number or high degree of oxygenation. They might be intermediately volatile or semi-volatile VOCs that were generally missed in most GC methods, for instance, traditional GC-FID/GC-MS analysis of offline samples. These I/SVOCs are found now important players in both ozone and secondary organic aerosol formation. The capability of the RT-PTR to show the signals might be one advantage. If semi-quantification for these compounds is possible, e.g., using an assumed k_ptr, would it be possible to have some rough estimation on the concentration levels of the less explored VOC species during field campaigns, e.g., the current one in this study in winter Shanghai?

We agree with the reviewer that the gray data points in Figure 2a might be intermediately volatile or semi-volatile VOCs (I/SVOCs). Iodide (I⁻) is generally used as reagent ion when measuring I/SVOCs with CIMS. Nevertheless, we can estimate the concentration levels of these organics during field campaigns using k_ptr (Sekimoto et al., 2017), which reads,

(Line 183-188) Although with a high uncertainty, we estimated the concentration levels of these less explored VOC species, potentially being intermediately volatile or semi-volatile VOCs (I/SVOCs), with assumed $k_{PTR}$. The average concentration of gray data points in Figure 2a except for reagent ions and PAN ($NO_2^+$) measured in this field campaign was 0.70 ppb, and the upper and lower quartiles were 0.57 ppb and 0.85 ppb, respectively. Note that the bulk signal measured by the RT-PTR is the sum of many isomeric compounds, while the estimate of the $k_{PTR}$ covers only a limited number of substances, and the calculation of the $k_{PTR}$ itself has an uncertainty of at least 20-50% (Sekimoto et al., 2017). In addition, the loss of I/SVOCs in the sampling tube is not considered.

**Comment 2**: I see in Table 2 that some N-containing species, such as acetonitrile, acrylonitrile, propanenitrile etc., are of Category I, which means that the confidence level for identification and quantification is quite high. Would it be good to further comment on the quantification of these species in the campaign, since these species could be biomass burning tracers and are of interest to many readers.

We thank the reviewer for this helpful suggestion. We have revised our manuscript accordingly, which reads,

(Line 318-321) In addition, a number of N-containing species, such as acetonitrile, acrylonitrile, propanenitrile etc., are of Category I, which means that the confidence level for their identification and quantification is quite high. The consistency of the RT-PTR and GC-PTR measurements of these

N-containing species is shown in the supplemental material (Fig. S7), indicating that these species can be reliably used as tracers for biomass burning (Coggon et al., 2016; Gouw et al., 2003).

**Comment 3**: L65: the protonated ion might be a quasi-molecular ion, instead of a molecular ion.

We have updated the term to "a protonated quasi-molecular ion" throughout the manuscript.

**Comment 4**: L69: "exists" to "exist"?

Thank you for identifying this mistake. This typo has been corrected.

**Comment 5**: L312-318: the two "dominated"'s in the first and last sentences look a bit contradictory. Maybe changing the first one to "mainly contributed by", while ethylbenzene was really dominating (say, if its contribution was >50%).

We thank the reviewer for this comment. We have revised our manuscript accordingly, which reads,

"Also, in line with previous studies (Coggon et al., 2023; Gouw et al., 2003b; Warneke et al., 2003), the RT-PTR $C_8H_{11}^+$ signals were mainly contributed by ethylbenzene and xylenes because the sum of ethylbenzene and xylenes explained more than 95% of $C_8H_{11}^+$ signals as shown in Fig. 4b. In addition, the $C_8H_{11}^+$ signal was dominated by xylenes, and only ~8% of the total signals was ethylbenzene."

**Reference**

Coggon, M. M., Veres, P. R., Yuan, B., Koss, A., Warneke, C., Gilman, J. B., Lerner, B. M., Peischl, J., Aikin, K. C., Stockwell, C. E., Hatch, L. E., Ryerson, T. B., Roberts, J. M., Yokelson, R. J., and Gouw, J. A. de: Emissions of nitrogen-containing organic compounds from the burning of herbaceous and arboraceous biomass: Fuel composition dependence and the variability of commonly used nitrile tracers, Geophys. Res. Lett., 43, 9903–9912, https://doi.org/10.1002/2016gl070562, 2016.

Gouw, J. A. de, Warneke, C., Parrish, D. D., Holloway, J. S., Trainer, M., and Fehsenfeld, F. C.: Emission sources and ocean uptake of acetonitrile ($CH_3CN$) in the atmosphere, J. Geophys. Res.: Atmos., 108, https://doi.org/10.1029/2002jd002897, 2003.

Sekimoto, K., Li, S.-M., Yuan, B., Koss, A., Coggon, M., Warneke, C., and Gouw, J. de: Calculation of the sensitivity of proton-transfer-reaction mass spectrometry (PTR-MS) for organic trace gases using molecular properties, Int J Mass Spectrom, 421, 71–94, https://doi.org/10.1016/j.ijms.2017.04.006, 2017.